



# Measurement report: The importance of biomass burning in light extinction and direct radiative effect of urban aerosol during the COVID-19 lockdown in China

Jie Tian[1,2], Qiyuan Wang[1,2,3], Huikun Liu[1], Yongyong Ma[4], Suixin Liu[1,2], Yong Zhang[1], Weikang Ran[1], Yongming Han[1,2,3], and Junji Cao[5]

[1]State Key Laboratory of Loess and Quaternary Geology, Institute of Earth Environment, Chinese Academy of Sciences, Xi'an 710061, China
[2]CAS Center for Excellence in Quaternary Science and Global Change, Xi'an 710061, China
[3]Guanzhong Plain Ecological Environment Change and Comprehensive Treatment National Observation and Research Station, Xi'an 710061, China
[4]Meteorological Institute of Shaanxi Province, Xi'an 710015, China
[5]Institute of Atmospheric Physics, Chinese Academy of Sciences, Beijing 100029, China

*Correspondence*: Qiyuan Wang (wangqy@ieecas.cn) and Junji Cao (jjcao@mail.iap.ac.cn)

**Abstract.** To mitigate climate change in China, a better understanding of optical properties of aerosol is required due to the complexity in emission sources. Here, an intensive real-time measurement was conducted in an urban area of China before and during the lockdown of Coronavirus Disease 2019 (COVID-19), to explore the impacts of anthropogenic activities on aerosol light extinction and direct radiative effect (DRE). The mean light extinction coefficient ($b_{ext}$) reduced from 774.7 $\pm$298.1 Mm$^{-1}$ during the normal period to 544.3 $\pm$179.4 Mm$^{-1}$ during the lockdown period. The generalized addictive model analysis indicated that the large decline of $b_{ext}$ (29.7%) was entirely attributed to the sharp reductions in anthropogenic emissions. Chemical calculation of $b_{ext}$ based on the ridge regression analysis showed that organic aerosol (OA) was the largest contributor to $b_{ext}$ in both periods (45.1–61.4%), and contributions of two oxygenated OAs to $b_{ext}$ increased by 3.0–14.6% during the lockdown. A hybrid environmental receptor model combining with chemical and optical variables identified six sources of $b_{ext}$. It was found that $b_{ext}$ from traffic-related emission, coal combustion, fugitive dust, nitrate plus secondary OA (SOA) source, and sulfate plus SOA source decreased by 21.4–97.9% in the lockdown, whereas $b_{ext}$ from biomass burning increased by 27.1% mainly driven by undiminished needs of residential cooking and heating. The atmospheric radiative transfer model was further used to illustrate that biomass burning instead of traffic-related emission became the largest positive effect (10.0 $\pm$ 10.9 W m$^{-2}$) on aerosol DRE in the atmosphere during the lockdown. Our study provides insights into aerosol $b_{ext}$ and DRE from anthropogenic sources, and the results implied the importance of biomass burning for tackling climate change in China in the future.





## 1 Introduction

The abrupt outbreak of Coronavirus Disease 2019 (COVID-19) since December of 2019 caused unprecedented economic and social disruption (Yao et al., 2020). Chinese government implemented the city lockdown and a series of strict restrictions on travel, transports, factories, and constructive activities for numerous cities in China to curb the virus
spread among humans. This provides a rare opportunity to investigate the impacts of anthropogenic activities on air pollution in China. Recent aerosol studies are conducted during the lockdown with a major focus on primary emissions and secondary formation, and most of them have revealed changes in aerosol compositions, sources, and processes under emission control measures (Le et al., 2020; Li et al., 2020; Wang et al., 2020a; Wang et al., 2020c; Zhao et al., 2020; Zheng et al., 2020). However, only a few studies are conducted to explore the link of chemical constituents in aerosol
with light absorption during the lockdown (Chen et al., 2020; Lin et al., 2021; Xu et al., 2020). The influences of reduced anthropogenic activities on the variations of aerosol optical properties and direct radiative effect (DRE) are less understood.

Atmospheric aerosols alter the radiative energy budget by directly scattering and absorbing solar and terrestrial radiation to affect the globe climate change (Bellouin et al., 2013; Yao et al., 2017). The spatiotemporal variations of aerosol
optical properties (e.g., light scattering coefficient ($b_{scat}$), light absorption coefficient ($b_{abs}$), light extinction coefficient ($b_{ext}$), and single scattering albedo (SSA)) that highly depended on their chemical compositions and sources (Malm and Hand, 2007; Tao et al., 2014), can result in uncertainties in estimating aerosol DRE (IPCC, 2013; Ma et al., 2012). Therefore, distinguishing chemical composition- and source-specific aerosol optical properties from a mixture of aerosols in the atmosphere would make a better understanding of the climate change during the COVID-19 lockdown.

The relationship between aerosol optical coefficients and chemical compositions can be built by the Interagency Monitoring of Protected Visual Environments algorithm and multiple linear regression (MLR) (Deng et al., 2016; Malm and Hand, 2007; Shen et al., 2014; Tao et al., 2014, 2015). However, previous studies often regarded organic aerosol (OA) as a whole light scattering component only. In reality, there are some OA components can absorb light, which is collectively termed as brown carbon (BrC) (Andreae and Gelencsér, 2006). The DRE caused by BrC has been reported
to be nonnegligible (e.g., 0.04 W m$^{-2}$ to 0.57 W m$^{-2}$) (Feng et al., 2013; Lin et al., 2014; Wang et al., 2014). Furthermore, the optical properties of OA can vary widely due to the complexity of OA components associated with primary sources, formation pathways, and aging processes (Laskin et al., 2015). For instance, primary OA (POA) from anthropogenic sources (e.g., biomass burning and coal combustion) usually has different mass scattering and absorption efficiencies (MSE and MAE) in the atmosphere compared to secondary OA (SOA) formed though photochemical or aqueous-phase
oxidations (Han et al., 2015; Qin et al., 2018). Therefore, investigating POA and SOA contributions to aerosol light scattering and absorption would reduce uncertainties in chemical apportionment of aerosol optical properties.



Previous studies have been conducted on the aerosol optical source apportionment. According to the multi-wavelength aethalometer measurement, the source of aerosol $b_{abs}$ can be investigated by exploiting the differences in absorption spectra of light-absorbing materials (Herich et al., 2011; Sandradewi et al., 2008; Zotter et al., 2017). In this method, the

aerosol absorption near ultraviolet and short-visible regions of the spectrum from biomass burning is assumed to be enhanced because of BrC emitted, compared to that from fossil fuel combustion (Kirchstetter et al., 2004; Tian et al., 2019). This makes it possible to derive their contributions to light absorption by using the specific source absorption Ångström exponent (AAE), but the so-called "aethalometer model" could not distinguish as many sources resolved by receptor models due to the similar optical properties of the aerosol sources (Saarikoski et al., 2021). In contrast, receptor

models can be utilized to resolve multiple optical source apportionment of aerosol. Several studies used a combination of the receptor model and MLR to indirectly identify sources of aerosol $b_{scat}$, $b_{abs}$, and $b_{ext}$ (Cao et al., 2012; Tian et al., 2020; Zhou et al., 2017). For example, Zhou et al. (2017) firstly used positive matrix factorization analysis to quantify the mass contributions of aerosol from secondary aerosol, biomass burning, traffic-related emissions, and coal burning based on the sole chemical species, and then the MLR was used to apportion the contribution of each source to $b_{scat}$ and

$b_{abs}$. In addition, recent studies have attempted to conduct direct optical source apportionment by combining aerosol chemical species with optical coefficients in one receptor model (Forello et al., 2019; Q. Wang et al., 2020b; Xie et al., 2019). This promising method can provide both chemical and optical profiles in each source to improve the performance of source identification, and may eliminate potential uncertainties caused by the indirect approach.

The Fenwei Plain is designated as the key regions of pollution treatment in "three-year action plan to fight air pollution"

implemented by the Chinese State Council in 2018. As one of megacities in this plain, Xi'an has been facing severe air pollution problem, especially in winter (Niu et al., 2016; Wang et al., 2015). Here, we conducted highly time-resolved aerosol $b_{scat}$ and $b_{abs}$ measurements in Xi'an before and during the city lockdown in China. The main objectives are to (1) characterize the changes of aerosol optical properties since COVID-19 lockdown; (2) quantify the contributions of individual chemical composition and specific source to $b_{ext}$; and (3) evaluate source-specific aerosol DRE based on a

radiative transfer model. This study provides insights into the response of aerosol $b_{ext}$ and DRE to anthropogenic emission sources, which is a scientific basis for making future emission control policies to deal with climate change in China.

## 2 Methodology

### 2.1 Sampling site and period

Intensive measurements of aerosol optical properties were conducted at an urban site located at Guanzhong Plain Ecological Environment Change and Comprehensive Treatment National Observation and Research Station, southwest



of Xi'an downtown (34°13' N, 108°52' E, Figure S1). All instruments were placed at the rooftop of an office building (~ 10 m above the ground) and approximately 30 m from the nearest traffic road. Detailed description of the sampling site can be found in Tian et al. (2021). In this study, the sampling campaign consisted of two distinct periods: normal period

(1 to 23 January, 2020) and COVID-19 lockdown period (27 January to 7 February, 2020). Three days of 24 to 26 January, 2020 were excluded due to the intensive influence of fireworks.

## 2.2 Measurements

### 2.2.1 Real-time measurements of $b_{scat}$ and $b_{abs}$

A single wavelength integrating nephelometer (Aurora 1000, Ecotech, Melbourne, Australia) was carried out to measure

aerosol $b_{scat}$ at a wavelength of 525 nm with 5-min time resolution. In the measurement volume, the ambient air sampled with a flow rate of 5 L min$^{-1}$ was illuminated by the light source, that only light scattered at scattering angles between 10 ° and 170 ° can reached the photomultiplier tube. Thereafter, $b_{scat}$ can be calculated by the proportion of the electrical signals produced by the photomultiplier tube. Span calibration was made using $CO_2$ to ensure accuracy of the instrument before sampling, and zero calibration were performed twice each day with particle-free air to subtract the Rayleigh

scattering. More detailed principles of the Aurora 1000 have been described in elsewhere (Chamberlain-Ward and Sharp, 2011).

Aerosol $b_{abs}$ at wavelengths of 370 nm, 470 nm, 520 nm, 590 nm, 660 nm, and 880 nm were measured by a newly developed Aethalometer (model AE33, Magee Scientific, Berkeley, CA, USA) with 1-min time resolution. Briefly, the model AE33 was the filter-based absorption photometer that simultaneously measured the light attenuation transmitted

through two parallel spots of the aerosol filter with 3.85 L min$^{-1}$ and 1.15 L min$^{-1}$, respectively. Based on "dual-spot" measurements, it used a real-time loading effect compensation algorithm to eliminate the nonlinear loading effect as increasing deposition amount of aerosol on the filter. Additionally, a factor of 2.14 was used in the model AE33 to automatically modify the quartz filter matrix scattering effect. A detailed description of this instrument can be found in Drinovec et al. (2015).

Both of the Aurora 1000 and model AE33 instruments equipped with a $PM_{2.5}$ cyclone separator in the sampling inlet to remove particles larger than 2.5 μm, and a Nafion® dryer (MD-700-24S, Perma Pure, Inc., Lakewood, NJ, USA) to retain particles (relative humidity < 40%) before entering these instruments. The amount of $b_{ext}$ in this study was defined as the sum of $b_{scat}$ at 525 nm and $b_{abs}$ at 520 nm.

### 2.2.2 Complementary data

A quadrupole aerosol chemical speciation monitor (Q-ACSM, Aerodyne Research Inc., Billerica, Massachusetts, USA) and a Xact 625 ambient metals monitor (Xact 625i, Cooper Environmental Services, Beaverton, OR, USA) were operated





to obtain chemical composition characteristics (Furger et al., 2020; Ng et al., 2011). The Q-ACSM measured concentrations of non-refractory species in $PM_1$ ($NO_3^-$, $SO_4^{2-}$, $NH_4^+$, $Cl^-$, and OA), and OA was further resolved into POA, less-, and more-oxidized oxygenated OA (LO-OOA and MO-OOA). Detail information on Q-ACSM data process
and source apportionment of OA can be found in our previous paper (Tian et al., 2021). The Xact 625i quantified hourly element concentrations through X-ray fluorescence analysis, including Si, K, Ca, Cr, Mn, Fe, Zn, As, Se, Ba, Hg, and Pb. Additionally, BC concentration was calculated using $b_{abs}$ at 880 nm (Kirchstetter et al., 2004). Online $PM_{2.5}$ and $NO_x$ concentrations were obtained from the Department of Ecology and Environment of Shaanxi Province. More detailed descriptions of these complementary data can be found in Table S1.

### 2.3 Meteorological conditions separation


A generalized addictive model (GAM) combined with integrated smoothness estimation was used to establish the relationship between $b_{ext}$ and several meteorological parameters as follows (Wood, 2004):

$$\ln b_{ext}(i) = \sum_{j=1}^{7} f_j\left(MP_j(i)\right) + \beta_0 + e_i \qquad (1)$$

where $b_{ext}(i)$ is the $b_{ext}$ in $Mm^{-1}$ averaged over the $i^{th}$ hour; $MP_j$ represents the $j^{th}$ meteorological parameter, such as wind
speed, wind direction, relative humidity, temperature, pressures, dew point, and planetary boundary layer height, where the data sources can be found in Table S1; $f$ corresponds to the smooth function describing the association between $b_{ext}$ and meteorological parameters; $\beta_0$ is the model intercept; and $e_i$ is the regression residuals which is assumed to be normally distributed.

Based on the R package "mgcv" (Wood, 2017), the whole campaign dataset was divided into three parts: a model data
(80% of data during the normal period) for establishing the $b_{ext}$ GAM, a test data (20% of data during the normal period) for verifying the accuracy of the model, and a forecast data (100% of data during the lockdown period) for estimating the contributions of meteorological conditions and emissions on $b_{ext}$ reduction.

### 2.4 Chemical calculation of $b_{scat}$ and $b_{abs}$

In view of POA and SOA with nonnegligible light scattering and absorbing abilities, the amount of $b_{scat}$ and $b_{abs}$
associated with individual chemical species can be estimated statistically using the ridge regression method:

$$b_{scat} = a_1[NH_4NO_3] + a_2[(NH_4)_2SO_4] + a_3[\text{fine soil}]$$
$$+ a_4[POA] + a_5[\text{LO-OOA}] + a_6[\text{MO-OOA}] + c_1 \qquad (2)$$

$$b_{abs} = b_1[BC] + b_2[POA] + b_3[\text{LO-OOA}] + b_4[\text{MO-OOA}] + c_2 \qquad (3)$$



where $b_{scat}$ and $b_{abs}$ are given in unit of $Mm^{-1}$; the bracket notation [] represents the specific chemical species
concentration in μg $m^{-3}$; the $a_i$ and $b_i$ ($i$ = 1–6) describe the MSE and MAE of each chemical species in unit of $m^2$ $g^{-1}$,
respectively; and $c_i$ ($i$ = 1 or 2) is the constant. In equation (2), the concentrations of [$NH_4NO_3$], [($NH_4$)$_2SO_4$], and [fine
soil] were calculated using 1.29×[$NO_3^-$], 1.35×[$SO_4^{2-}$], and [Fe]/0.032, respectively (Chow et al., 2015; CNEMC, 1990).
In equation (3), $b_1$ was calculated by absorption Ångström exponent method, and the detailed description can be seen in
Text S1.

**2.5 Hybrid environmental receptor model (HERM) for source apportionment**

The source apportionment of $b_{ext}$ was performed with HERM which is a newly developed bilinear model (Chen and Cao,
2018). Briefly, the HERM solves non-negative matrices of unknown factor profiles and contributions with a pre-set
number of factors K by iteratively minimizing the object function Q defined as follows:

$$Q = \sum_{j=1}^{J} \sum_{i=1}^{I} \frac{\left(x_{ij} - \sum_{k=1}^{K} g_{ik} f_{kj}\right)^2}{\sigma_{x_{ij}}^2 + \sum_{k=1}^{K}\left(g_{ik}^2 \sigma_{f_{kj}}^2 + \delta_{ik}\sigma_{x_{ij}}^2\right)} \tag{4}$$

where I, J, and K are the number of samples, aerosol variables, and factors, respectively; the indices of $i$, $j$, $k$ represent
the sample, aerosol variable, and factor, respectively; $x_{ij}$ is the measured ambient data spectral matrix; $f_{kj}$ is the factor
profile matrix; $g_{ik}$ is factor contribution matrix; $\sigma_{x_{ij}}$ and $\sigma_{f_{kj}}$ represent the error in measured ambient data and variability
in constrained factor profile, respectively; $\delta_{ik}$ is set to 0 or 1 depending on whether the $k^{th}$ factor profile is constrained or
unconstrained, respectively.

In this study, both chemical species ($PM_{2.5}$, $NO_3^-$, $SO_4^{2-}$, $NH_4^+$, $Cl^-$, BC, POA, LO-OOA, MO-OOA, Si, K, Ca, Cr, Mn,
Fe, Zn, As, Se, Ba, Hg, and Pb in μg $m^{-3}$) and optical variables ($b_{scat}$ and $b_{abs}$ in $Mm^{-1}$) were used as input data for the
HERM analysis. The uncertainties of hourly ambient data except elements were introduced by the standard deviation of
samples with higher time resolution (< 1-hour); the uncertainty of the element was estimated using its concentration, the
default analytical relative error (10%) (Rai et al., 2020), and method detection limit (MDL) (Norris et al., 2014) (Text
S2). All input variables were classified as strong due to the high signal-to-noise (SNR > 2). Here, the HERM had
predetermined: (1) the $i^{th}$ sample was excluded from source apportionment when missing values occurred in variables;
(2) $PM_{2.5}$ value in factor profile was set to unity as a reference standard for both chemical and optical variables.

A range of factor numbers from 2 to 8 was selected to run in the HERM software with completely unconstrained factor
profiles, and diagnostic plots are detailed in the supplementary material (Text S3 and Figure S2–S7). The 6-factor
solution without mixed source was found to be the optimal solution based on multiple criteria including (1) variations in
$Q/Q_{exp}$ that can be used as a metric for choosing the best number of resolved factors (Ulbrich et al., 2009); (2) physical





meaningfulness of distinct factor profiles and explained variations (EV) of variables; (3) agreement between the measured and modeled values; and (4) good correlations with external and internal tracers. Detailed information on the final selected factor profiles and contributions are presented in Section 3.4.

**2.6 DRE calculations**

The Santa Barbara DISORT Atmospheric Radiative Transfer (SBDART) developed by Institute for Computational Earth System Science, University of California was utilized to estimate the source-specific aerosol DRE. It can calculate the downwelling and upwelling radiative flux ($F_{down}$ and $F_{up}$), in which difference indicates the net radiative flux ($\Delta F = F_{down} - F_{up}$). A detailed description of the SBDART can be found in Ricchiazzi et al. (1998). Based on the optical source

apportionment results, the SBDART model input values of aerosol optical depth, SSA, asymmetry factor, and optical coefficients were retrieved using the Optical Properties of Aerosol and Cloud (OPAC) model (Hess et al., 1998). The aerosol DRE can be calculated as follows:

$$DRE_{atmosphere} = DRE_{top} - DRE_{surface} \tag{5}$$

$$DRE_{top} = \Delta F_{top}(\text{with aerosol}) - \Delta F_{top}(\text{without aerosol}) \tag{6}$$

$$DRE_{surface} = \Delta F_{surface}(\text{with aerosol}) - \Delta F_{surface}(\text{without aerosol}) \tag{7}$$

where the indices of atmosphere, top, and surface indicate the DRE in the atmosphere, at the top of the atmosphere, and the earth's surface, respectively; $\Delta F$(with aerosol) and $\Delta F$(without aerosol) represent the net radiative flux with and without aerosol, respectively.

**3 Results and discussion**

**3.1 General descriptions of aerosol optical properties**

The temporal variations of hourly mean $b_{scat}$, $b_{abs}$, $b_{ext}$, and SSA together with PM$_{2.5}$ mass concentrations for the entire sampling period are depicted in Figure S8, while a statistics summary of optical and chemical parameters during the normal and COVID-19 lockdown periods is shown in Table 1. The optical coefficients decreased dramatically in accord with the significant reduction of PM$_{2.5}$ since stringent control measures on emission sources implemented during the

lockdown period (Tian et al., 2021; Zheng et al., 2020). The mean values of $b_{scat}$, $b_{abs}$, and $b_{ext}$ during the normal period were 688.1 ±261.4 Mm$^{-1}$, 86.6 ±43.0 Mm$^{-1}$, and 774.7 ±298.1 Mm$^{-1}$, respectively, which is consistent with the values (657.4 ±436.9 Mm$^{-1}$, 104.0 ±69.6 Mm$^{-1}$, and 761.4 ±506.5 Mm$^{-1}$) reported previously in winter of 2009 in Xi'an (Cao et al., 2012), even though a series of nationwide air quality standards and long-term pollution control policies have been





implemented in the 74 major cities since 2013 (Zheng et al., 2018). Comparatively, the kind of control measures aiming
to curb the outbreaks did not last long, but it was unprecedentedly strictest in China. The large decreases (27.6–47.0%)
were found in $b_{scat}$, $b_{abs}$, and $b_{ext}$ in the lockdown (498.4 $\pm$ 159.0 Mm$^{-1}$, 45.9 $\pm$ 22.9 Mm$^{-1}$, and 544.3 $\pm$ 179.4 Mm$^{-1}$,
respectively), providing insights into the role of anthropogenic emissions on aerosol optical properties.

The SSA defined as the ratio of $b_{scat}$ to $b_{ext}$ increased from 0.89 $\pm$ 0.03 during the normal period to 0.92 $\pm$ 0.02 during
the lockdown period. As presented in Figure S9a and b, SSA showed linear increases with the mass fractions of secondary
inorganic aerosol (SIA = NH$_4$NO$_3$ + (NH$_4$)$_2$SO$_4$) to PM$_{2.5}$ (R$^2$ = 0.83–0.84) and SOA (SOA = LO-OOA + MO-OOA) to
OA (R$^2$ = 0.94–0.99), indicating an enhanced role of secondary formation in the lockdown. In addition, the correlations
of SSA and the ratio of LO-OOA to MO-OOA were established to reveal a more complex influence of SOA on SSA
(Figure S9c), which showed obviously negative relationships (R$^2$ = 0.69–0.79). It indicated that SSA can be impacted
by the degree of oxidation on aerosol, and higher scattering and lower absorption abilities are usually found for more
oxidized OA (Han et al., 2015; Lee et al., 2014).

### 3.2 Effects of emission reduction and meteorological conditions on reduced $b_{ext}$

Figure 1 shows the time series of the measured and GAM-predicted $b_{ext}$ for the model data, test data, and forecast data.
As shown in Table S2 and S3 the constructed GAM with adjusted R$^2$ value (0.569) can explain 56.9% of the variation
in $b_{ext}$ after incorporating the nonlinear relationships between optical and meteorological parameters. Independent
smoothed meteorological variables of the model were statistically significant by according to $p$ values ($< 0.05$) from F
test. Concurvity indices between each independent smoothed parameter were within 0.5, indicating there was no serious
multicollinearity (Schimek, 2009).

Before applying the constructed GAM to predict the $b_{ext}$ during the lockdown period, the cross-validation test was used
to evaluate the model. For the test data (20% of data during the normal period), the R$^2$ value of the linear regression and
index of agreement (IOA) (Wu et al., 2018) between the measured and GAM-predicted $b_{ext}$ was 0.80 and 0.91,
respectively, suggesting a good performance of the constructed GAM. Therefore, the difference between the measured
and GAM-predicted $b_{ext}$ in the lockdown can be attributed to emission reduction through the implementation of stringent
control measures on emission sources. The emission reduction decreased $b_{ext}$ by 294.6 Mm$^{-1}$ during the lockdown period,
higher than the decline of measured $b_{ext}$ (230.4 Mm$^{-1}$) from normal to lockdown periods. It is indicated that the
meteorological conditions enhanced $b_{ext}$ by 64.2 Mm$^{-1}$ during the lockdown period, further reflecting the effective control
of anthropogenic emissions.



### 3.3 Contribution of chemical components to $b_{ext}$

Table 2 presents the estimated MSE and MAE of individual chemical component during the normal and lockdown periods. The MSEs of NH$_4$NO$_3$ (3.74 ±0.18 m$^2$ g$^{-1}$) and (NH$_4$)$_2$SO$_4$ (7.35 ±0.25 m$^2$ g$^{-1}$) during the normal period were higher than those (3.23 ±0.18 m$^2$ g$^{-1}$ and 4.78 ±0.35 m$^2$ g$^{-1}$) during the lockdown period. This may be explained by the higher mass loadings and peak diameters of aerosol without control measures (Cheng et al., 2015; Tao et al., 2015). The MAE of BC decreased from 15.00 m$^2$ g$^{-1}$ to 13.27 m$^2$ g$^{-1}$ that related to the decline of AAE of BC (Text S1). The MSEs and MAEs of OA factors varied widely, from 3.48 m$^2$ g$^{-1}$ to 12.89 m$^2$ g$^{-1}$ and from 0.25 m$^2$ g$^{-1}$ to 0.59 m$^2$ g$^{-1}$, respectively, due to the complex chemical variability of OA constituents (Hallquist et al., 2009; Moise et al., 2015). The scattering ability of OA increased with oxidation level (from POA to MO-OOA) (Cappa et al., 2011; Flores et al., 2014); however, the dependence on oxidation level of OA MAEs presented more complex trend. LO-OOA had the higher MAE values than those of POA, indicating more BrC chromophores with stronger light-absorbing capacity formed under less-oxidized condition (Zhang et al., 2020). Additionally, the effect of photo-bleaching in the atmosphere that can weaken the light absorption ability of BrC that resulted in the reduction of MO-OOA MAEs (Wang et al., 2021).

Chemical calculation of $b_{ext}$ was confirmed to be a reasonable estimation of aerosol optical coefficients by using chemical components data (Figure S10 and S11). As shown in Figure 2, OA (POA + LO-OOA + MO-OOA) was the largest contributor to $b_{ext}$ in both periods, accounting for 45.1–61.4%, followed by NH$_4$NO$_3$ (16.5–24.1%), BC (9.3–13.1%), (NH$_4$)$_2$SO$_4$ (7.9–11.2%), and fine soil (4.9–6.5%). This result was different from previous findings that SIA was often the largest contributor to $b_{ext}$ in China, such as Beijing (46–54%) (Han et al., 2015), Chengdu (43%) (Tao et al., 2014), Nanjing (53%) (Shen et al., 2014), and Xi'an (63%) (Cao et al., 2012), highlighting the dominant role of organic matters in aerosol light extinction in Xi'an today. Compared to the normal period, the contributions of NH$_4$NO$_3$, (NH$_4$)$_2$SO$_4$, fine soil, and BC, and POA to $b_{ext}$ decreased by 1.3–7.6% in the lockdown, whereas contributions of two SOAs to $b_{ext}$ increased by 3.0–14.6%. On the one hand, the mass concentrations of LO-OOA and MO-OOA decreased by 20.9–34.7% from normal to lockdown periods, lower than those of other chemical species (35.8–72.5%); On the other hand, both of SOAs MSEs and MAEs showed higher values during the lockdown period, especially MO-OOA. The combination of effects eventually led to an enhanced role of SOA in light extinction during the lockdown.

### 3.4 Contribution of sources to $b_{ext}$

The 6-factor solution was selected to be the optimal solution, which can adequately account for the variability in aerosol $b_{ext}$ (Figure S12). Six sources were determined by the HERM analysis, consisting of traffic-related emission, biomass burning, coal combustion, fugitive dust, nitrate plus SOA source, and sulfate plus SOA source. Details about their characteristics are presented in Figure 3. The first source identified as traffic-related emission was characterized by high





EV values of Cr (77%), Mn (53%), Fe (36%), and Zn (39%), which can be released from lubricating oils, fuel additives, and brake and tire wear (Ålander et al., 2005; Geivanidis et al., 2003; Tao et al., 2017; Zhang et al., 2013). Moderate contributions of POA (26%) and BC (28%) were commonly regarded as species of diesel and gasoline engine exhaust (Chow et al., 2004; Liu et al., 2017). Additionally, the temporal variations in $b_{ext}$ from this source correlated well with $NO_x$ ($R^2 = 0.72$), suggesting an association with motor vehicle emissions (Huang et al., 2017; Li et al., 2017). The second source with high EV values of POA (45%), LO-OOA (41%), BC (32%), Cl (34%), and K (41%) was judged to biomass burning. K were regarded as an excellent tracer of biomass burning (Li et al., 2007; Ni et al., 2017), and good correlations were also found between $b_{ext}$ from biomass burning and K ($R^2 = 0.64$). Previous studies have shown that POA from biomass burning can be rapidly oxidized in the atmosphere (Cubison et al., 2011), therefore, the abundant LO-OOA observed in this source might be indicative of aged biomass-burning aerosol (Crippa et al., 2013; Kim et al., 2017; Xu et al., 2015). The third source, coal combustion, was characterized by high EV values of Cl (42%), As (38%), Se (46%), and Pb (25%). Of these elements, As and Se had been found to be enriched in coals (Tian et al., 2013), which were reliable indicators for coal combustion (Tan et al., 2017; Yu et al., 2019); and Pb was found to possibly emitted from coal combustion in Xi'an (Xu et al., 2012). The fourth source was defined as fugitive dust due to significant EV values of Si (92%), Ca (63%), and Fe (31%), which were the dominant chemical species in natural and construction dust profiles (Liu et al., 2017; Zhao et al., 2006). Two secondary sources were resolved in our study as nitrate plus SOA source with high EV values of $NO_3^-$ (42%), $NH_4^+$ (33%), and MO-OOA (34%) and sulfate plus SOA source with high EV values of $SO_4^{2-}$ (58%) and MO-OOA (39%), respectively. Since $SO_2$ oxidation to sulfate need long time (e.g., 1 week) at the typical atmospheric level of OH radicals, $SO_4^{2-}$ was likely associated with regional source, while $NO_3^-$ was often formed more locally due to the intense $NO_x$ emissions in China (Zhang et al., 2015; Zheng et al., 2014). The defined nitrate and sulfate plus SOA sources appeared to have stronger associations with local and regional processes, respectively.

As shown in Figure 4, the average $b_{ext}$ from traffic-related emission, coal combustion and fugitive dust decreased from 77.3 ±46.8 Mm$^{-1}$, 73.6 ±60.9 Mm$^{-1}$, and 93.3 ±82.7 Mm$^{-1}$ during the normal period to 1.7 ±4.0 Mm$^{-1}$, 38.5 ±34.5 Mm$^{-1}$, and 30.8 ±24.4 Mm$^{-1}$ during the lockdown period, respectively, which can be explained by traffic restriction, closure of industries and stopping construction activities. $b_{ext}$ from traffic-related emission with the largest reduction (97.9%) emphasized the effectiveness of controlling private gasoline cars and commercial and construction diesel trucks in the lockdown (Wang et al., 2020c). For two secondary sources, though previous studies reported the enhancement of secondary aerosol formation efficiencies as the increase of atmospheric oxidation capacity in the lockdown (Huang et al., 2020; Le et al., 2020; Tian et al., 2021), the decreases in gas and organic precursors (e.g., $NO_2$, $SO_2$, and VOCs) led to the 47.5% and 21.4% reductions of $b_{ext}$ from sources of nitrate plus SOA and sulfate plus SOA, respectively. That is, the enhanced secondary aerosol cannot offset the primary emission reduction in Xi'an, confirming that reducing anthropogenic primary emissions is still the most effective treatment of aerosol pollution.



By contrast, the average $b_{ext}$ from biomass burning during the lockdown period (215.4 ±163.9 Mm$^{-1}$) was higher than

that during the normal period (169.4 ±196.9 Mm$^{-1}$). The government didn't strengthen the past control policies that

forbade biomass burning in the lockdown. Moreover, strict controls were enforced on the movements of people, even in

the countryside, possibly resulting more consumption of biomass for cooking and heating. As shown in Figure 5, the

rising stages of PM$_{2.5}$ during the lockdown period were all accompanied by the increase in $b_{ext}$ from biomass burning,

accounting for 46.4–55.6% of the total $b_{ext}$. Take the rising stage of PM$_{2.5}$ from 13:00 30 to 7:00 31 January as an example,

$b_{ext}$ from POA and LO-OOA increased rapidly at rates of 8.6 Mm$^{-1}$ hour$^{-1}$ and 8.2 Mm$^{-1}$ hour$^{-1}$, respectively.

Correspondingly, $b_{ext}$ from biomass burning showed the fastest rise (26.0 Mm$^{-1}$ hour$^{-1}$) in all primary source, that led to

biomass burning became the most important source to $b_{ext}$ (36.7%) in the lockdown (Figure 4). Hence, additional actions

and investigations on biomass burning emissions would be taken into consideration.

### 3.5 Impacts of COVID-19 lockdown on aerosol DRE

Figure 6 shows the range of source-specific aerosol DRE$_{top}$, DRE$_{surface}$, and DRE$_{atmosphere}$ during the normal and lockdown

periods. For all sources, the aerosol DRE$_{atmosphere}$ values in both periods were positive, producing net warming effects in

the atmosphere. The mean aerosol DRE$_{atmosphere}$ reduced from 31.0 ±23.2 W m$^{-2}$ before the lockdown to 14.1 ±11.5 W

m$^{-2}$ in the lockdown, with a reduction of 54.5%.

With regard to the contributions of specific sources on the DRE$_{atmosphere}$, traffic-related emission had the largest positive

effect on DRE$_{atmosphere}$ during the normal period, with the value of 13.3 ±9.2 W m$^{-2}$, followed by biomass burning (8.4

±13.0 W m$^{-2}$), coal combustion (7.8 ±7.2 W m$^{-2}$), sulfate plus SOA source (1.7 ±3.0 W m$^{-2}$), and fugitive dust (1.1 ±

2.4 W m$^{-2}$). Nitrate and plus SOA source presented the negative value of DRE$_{atmosphere}$ (-1.2 ±0.7 W m$^{-2}$), suggesting the

cooling effect in the atmosphere. Due to the strictest traffic restrictions implemented, the DRE$_{atmosphere}$ from traffic-

related emission (0.4 ±1.0 W m$^{-2}$) showed the significant reduction (97.0%) in the lockdown. However, the DRE$_{atmosphere}$

from biomass burning increased to 10.0 ±10.9 W m$^{-2}$, indicating that biomass burning was not effectively controlled

during the lockdown period. The other four sources contributed relatively small amounts of DRE$_{atmosphere}$; that is 4.5 ±

4.5 W m$^{-2}$ for coal combustion, -0.3 ±0.8 W m$^{-2}$ for fugitive dust, -1.4 ±0.8 W m$^{-2}$ for nitrate plus SOA source, and 1.0

±1.8 W m$^{-2}$ for sulfate plus SOA source. The results indicated that the control measures on traffic in the lockdown were

highly effective for mitigating the effects of climate change in short-term, while future emission control policies should

consider the importance of biomass burning to tackle climate change in China.





# 4 Conclusion

This study conducted an intensive real-time measurement campaign in an urban city of China before and during the lockdown of Coronavirus Disease 2019 to investigate the impacts of anthropogenic emissions on aerosol optical properties and direct radiative effect (DRE). Decreases in light scattering coefficient ($b_{scat}$), light absorption coefficient
($b_{abs}$), and light extinction coefficient ($b_{ext}$) were observed in the lockdown with reductions of 27.6–47.0%, in accord with the decline of $PM_{2.5}$ under strict emission control measures. Single scattering albedo during the lockdown period (0.92 ± 0.02) was higher than that during the normal period (0.89 ± 0.03), suggesting an enhanced role of secondary formation in the lockdown. The generalized addictive model analysis showed that meteorological conditions enhanced $b_{ext}$ by 64.2 $Mm^{-1}$ during the lockdown period, thus, the dramatical reduction of $b_{ext}$ was totally credited to anthropogenic
emission reductions.

The relationship between $b_{ext}$ and chemical components was established based on the ridge regression analysis. Using the estimated mass scattering and absorption efficiencies (MSEs and MAEs) of chemical components, OA including primary OA, less-, and more-oxidized oxygenated OA was found to be the largest contributor (45.1–61.4%) to $b_{ext}$ before and during the lockdown period, followed by $NH_4NO_3$ (16.5–24.1%), BC (9.3–13.1%), $(NH_4)_2SO_4$ (7.9–11.2%), and
fine soil (4.9–6.5%). Particularly, secondary OA played an increasingly important part in light extinction during the lockdown when contributions of two oxygenated OAs to $b_{ext}$ increased by 3.0–14.6%.

A hybrid environmental receptor model coupled with chemical and optical variables was utilized to carried out optical source apportionment. Six sources of $b_{ext}$ were resolved, including traffic-related emission, biomass burning, coal combustion, fugitive dust, nitrate plus SOA source, and sulfate plus SOA source. Most of sources showed reductions of
$b_{ext}$ (21.4–97.9%) during the lockdown, confirming the effectiveness of reducing anthropogenic primary emissions for treating aerosol pollution. $b_{ext}$ from traffic-related emission had the most evident decrement (97.9%), whereas that from biomass burning increased by 27.1% during the lockdown due to the undiminished needs of residential cooking and heating in winter.

The atmospheric radiative transfer further illustrated that aerosol produced net warming effects (14.1–31.0 W $m^{-2}$) in the
atmosphere during the normal and lockdown periods. Biomass burning instead of traffic-related emission became the largest positive effect (10.0 ± 10.9 W $m^{-2}$) on aerosol DRE in the atmosphere in the lockdown. The results implied that reducing biomass burning would be another direct and effective way on climate change mitigation besides traffic restriction, therefore, Chinese government should further tighten the policy on controlling biomass burning in the future.



*Data availability.* Data used to support the findings in this study are archived at the Institute of Earth Environment,
Chinese Academy of Sciences, and are publicly available at https://doi.org/10.5281/zenodo.5739349.

*Competing interests.* The authors declare that they have no conflict of interest.

*Author contributions.* QW, YH, and JC designed the campaign. SL, YZ, and WR conducted field measurements. JT, HL,
and YM made data analysis and interpretation. JT wrote the paper with contributions from all co-authors.

*Acknowledgments.* The authors are grateful to the staff from Guanzhong Plain, Eco-environmental Change and
Comprehensive Treatment, National Observation and Research Station for their assistance with field sampling.

*Financial support.* This research was jointly supported by the Key Research and Development Program of Shaanxi
Province (grant no. 2018-ZDXM3-01), the Strategic Priority Research Program of Chinese Academy of Sciences (grant
no. XDB40000000), the West Light Foundation of the Chinese Academy of Sciences (grant no. XAB2019B05), the
Sino-Swiss Cooperation on Air Pollution Source Apportionment for Better Air (grant no. 7F-09802.01.02), and the
Youth Innovation Promotion Association of the Chinese Academy of Sciences (grant no. 2019402).

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





**Table 1.** Summary of optical coefficients and chemical species in Xi'an observed in the entire campaign, normal period (1 to 23 January), and COVID-19 lockdown period (27 January to 7 February).

| Parameters[*] | Entire campaign | Normal period | COVID-19 lockdown period | Change ratio[**] |
|---|---|---|---|---|
| Optical coefficients | | | | |
| $b_{scat}$ | 623.2 ±248.3 | 688.1 ±261.4 | 498.4 ±159.0 | 27.6% |
| $b_{abs}$ | 72.6 ±42.1 | 86.6 ±43.0 | 45.9 ±22.9 | 47.0% |
| $b_{ext}$ | 695.8 ±285.3 | 774.7 ±298.1 | 544.3 ±179.4 | 29.7% |
| SSA | 0.90 ±0.03 | 0.89 ±0.03 | 0.92 ±0.02 | -3.2% |
| | | | | |
| Chemical species | | | | |
| PM$_{2.5}$ | 116.4 ±56.3 | 134.4 ±56.9 | 81.8 ±34.9 | 39.1% |
| NH$_4$NO$_3$ | 33.1 ±17.3 | 40.2 ±16.4 | 19.5 ±8.8 | 51.6% |
| (NH4)$_2$SO$_4$ | 8.3 ±4.6 | 9.5 ±4.9 | 5.9 ±2.5 | 38.1% |
| fine soil | 11.8 ±8.0 | 15.8 ±7.2 | 4.3 ±1.9 | 72.5% |
| BC | 4.4 ±2.6 | 5.4 ±2.6 | 2.7 ±1.3 | 50.6% |
| POA | 18.3 ±12.4 | 20.9 ±12.7 | 13.4 ±10.1 | 35.8% |
| LO-OOA | 7.6 ±5.8 | 8.6 ±6.4 | 5.6 ±3.7 | 34.7% |
| MO-OOA | 11.1 ±4.5 | 12.0 ±4.8 | 9.5 ±3.3 | 20.9% |

[*]The units for $b_{scat}$, $b_{abs}$, $b_{ext}$ are Mm$^{-1}$; SSA is dimensionless; The units of chemical species are µg m$^{-3}$.

[**]Change ratio = ([Normal period] − [COVID-19 lockdown period])/[Normal period].



**Table 2.** Estimated MSEs and MAEs ($m^2$ $g^{-1}$) of individual chemical components during normal and COVID-19 lockdown periods.

| Components | Normal period | | COVID-19 lockdown period | |
|:---:|:---:|:---:|:---:|:---:|
| | MSE | MAE | MSE | MAE |
| $NH_4NO_3$ | 3.74 ±0.18 | | 3.23 ±0.18 | |
| $(NH_4)_2SO_4$ | 7.35 ±0.25 | | 4.78 ±0.35 | |
| fine soil | 2.46 ±0.35 | | 3.39 ±0.79 | |
| BC | | 15.00 | | 13.27 |
| POA | 3.90 ±0.18 | 0.25 ±0.01 | 3.48 ±0.16 | 0.29 ±0.01 |
| LO-OOA | 8.62 ±0.27 | 0.27 ±0.02 | 9.87 ±0.35 | 0.59 ±0.03 |
| MO-OOA | 9.87 ±0.45 | / | 12.89 ±0.55 | 0.31 ±0.04 |

[*]MAE of MO-OOA during the normal period was negative (near zero) and not listed in the table.





**Figure captions:**

**Figure 1.** Time series of the measured and GAM-predicted light extinction coefficient ($b_{ext}$) for the model data, test data, and forecast data.

**Figure 2.** Contributions of $NH_4NO_3$, $(NH_4)_2SO_4$, fine soil, BC, POA, LO-OOA, and MO-OOA to the reconstructed chemical light extinction coefficient ($b_{ext}$) during the normal and COVID-19 lockdown periods.

**Figure 3.** (a) Profiles and (b) time series plots of the resolved source factors in the 6-factor solution, including traffic-related emission, biomass burning, coal combustion, fugitive dust, nitrate plus SOA source, and sulfate plus SOA source. The columns in each factor is the profile that displaying relative relation of absolute values of variables. The red dot represents the explained variation of species for different factors. The corresponding time trends of chemical tracers are also shown.

**Figure 4.** Contributions of six resolved sources to the modeled source light extinction coefficient ($b_{ext}$) during the normal and COVID-19 lockdown periods, including traffic-related emission, biomass burning, coal combustion, fugitive dust, nitrate plus SOA source, and sulfate plus SOA source.

**Figure 5.** Time series of $PM_{2.5}$ mass concentration, the light extinction coefficient ($b_{ext}$) of chemical species, and the $b_{ext}$ from six resolved sources during the lockdown period. Pie charts depicting the average fractional contributions of 665 chemical species and sources to $b_{ext}$ during the $PM_{2.5}$ rising stages, which were marked in light gray.

**Figure 6.** Direct radiative effect (DRE) of aerosol from traffic-related emission, biomass burning, coal combustion, fugitive dust, nitrate plus SOA source, and sulfate plus SOA source at the earth's surface, the top of the atmosphere, and in the atmosphere during the normal (a) and COVID-19 lockdown (b) periods.



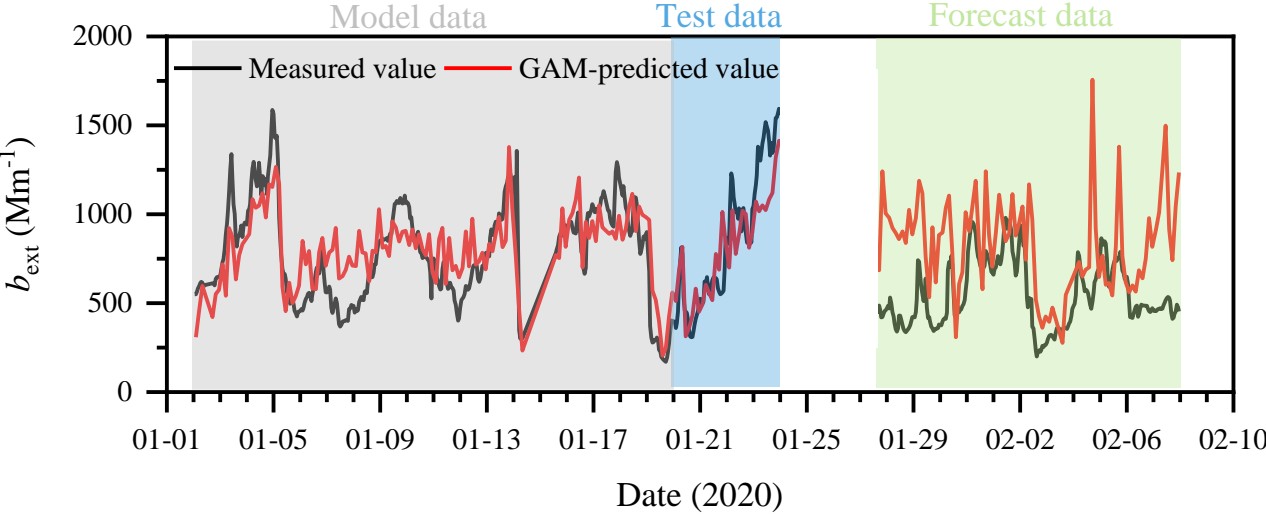

**Figure 1.** Time series of the measured and GAM-predicted light extinction coefficient ($b_{ext}$) for the model data, test data, and forecast data.





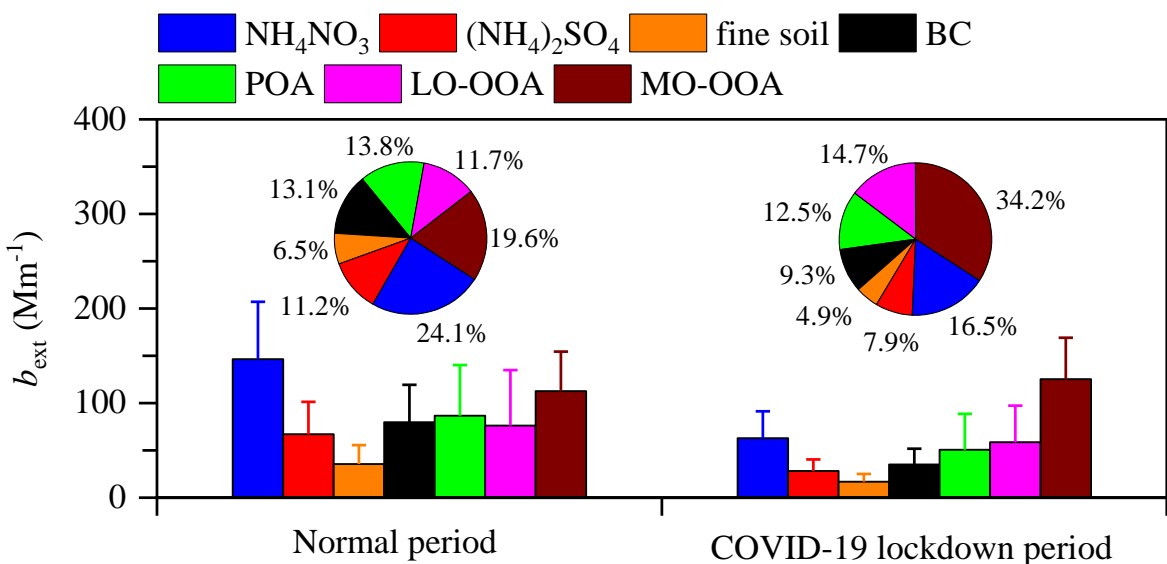

**Figure 2.** Contributions of $NH_4NO_3$, $(NH_4)_2SO_4$, fine soil, BC, POA, LO-OOA, and MO-OOA to the reconstructed chemical light extinction coefficient ($b_{ext}$) during the normal and COVID-19 lockdown periods.



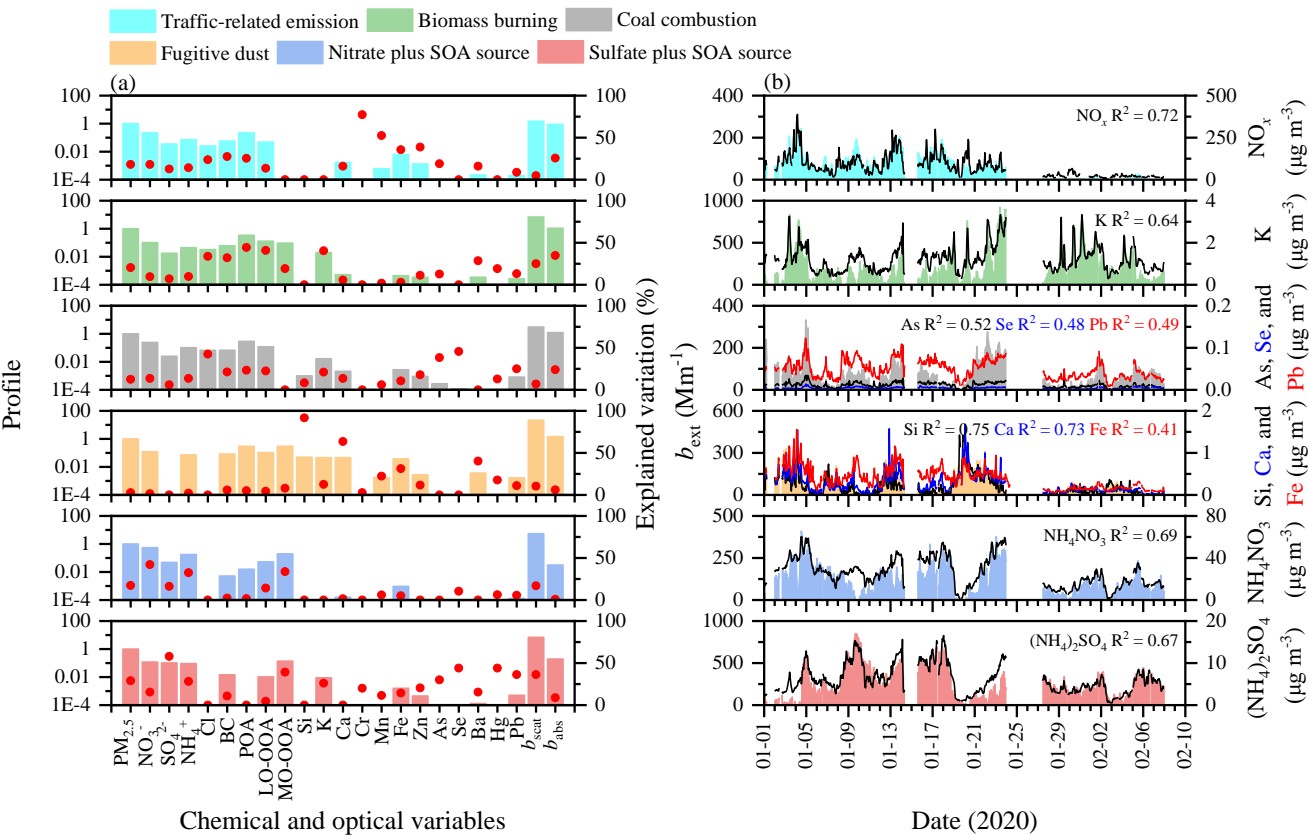

**Figure 3.** (a) Profiles and (b) time series plots of the resolved source factors in the 6-factor solution, including traffic-
related emission, biomass burning, coal combustion, fugitive dust, nitrate plus SOA source, and sulfate plus SOA source.
The columns in each factor is the profile that displaying relative relation of absolute values of variables. The red dot
represents the explained variation of species for different factors. The corresponding time trends of chemical tracers are
also shown.





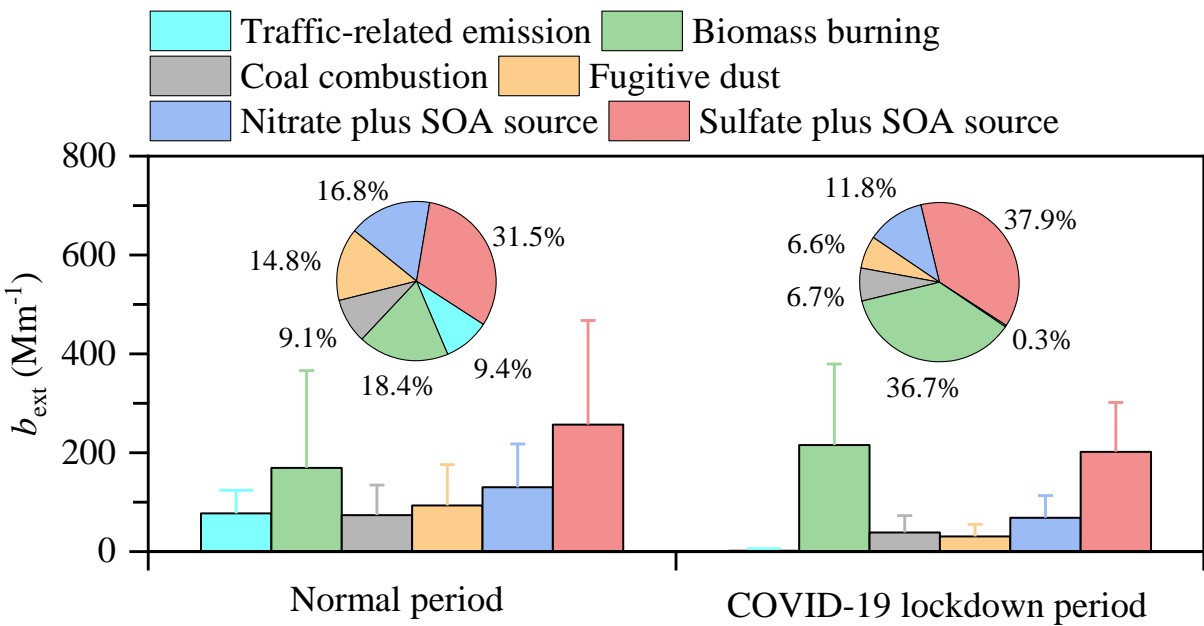

**Figure 4.** Contributions of six resolved sources to the modeled source light extinction coefficient ($b_{ext}$) during the normal and COVID-19 lockdown periods, including traffic-related emission, biomass burning, coal combustion, fugitive dust, nitrate plus SOA source, and sulfate plus SOA source.



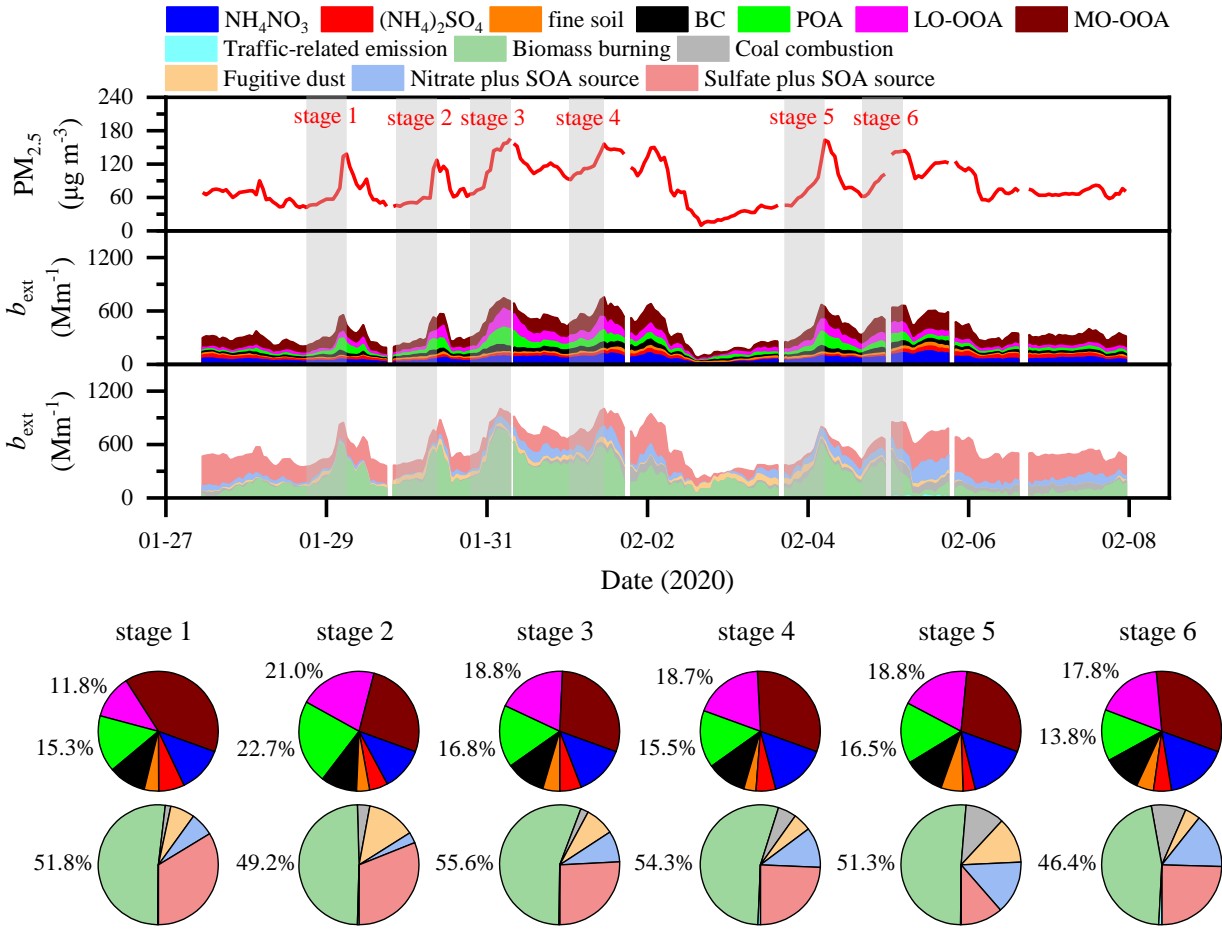

**Figure 5.** Time series of PM$_{2.5}$ mass concentration, the light extinction coefficient ($b_{ext}$) of chemical species, and the $b_{ext}$ from six resolved sources during the lockdown period. Pie charts depicting the average fractional contributions of chemical species and sources to $b_{ext}$ during the PM$_{2.5}$ rising stages, which were marked in light gray.





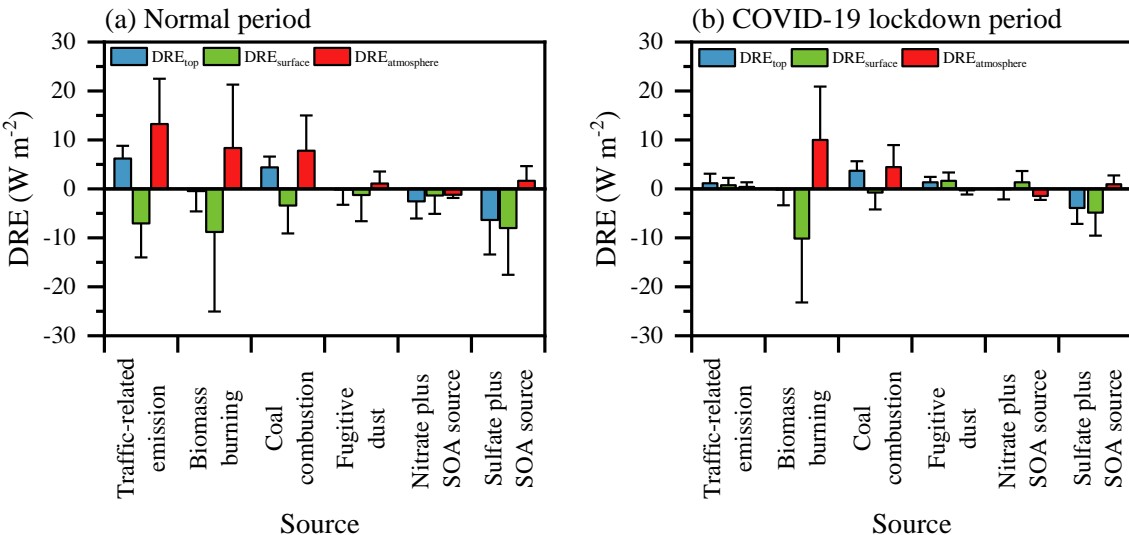

695

**Figure 6.** Direct radiative effect (DRE) of aerosol from traffic-related emission, biomass burning, coal combustion, fugitive dust, nitrate plus SOA source, and sulfate plus SOA source at the earth's surface, the top of the atmosphere, and in the atmosphere during the normal (a) and COVID-19 lockdown (b) periods.