# Peer review of "Measurement report: The importance of biomass burning in light extinction and direct radiative effect of urban aerosol during the COVID-19 lockdown in Xi'an, China"

_Atmospheric Chemistry and Physics, 2021_

## Author Response (AR1)

February 9, 2022

Dear editor,

Thank you for providing us the opportunity to revise and improve our manuscript entitled "Measurement report: The importance of biomass burning in light extinction and direct radiative effect of urban aerosol during the COVID-19 lockdown in China" by Tian et al. We are also grateful for the valuable comments and suggestions raised by reviewers.

We have made the changes suggested by reviewers and have outlined those changes in detail below. The modifications can be tracked in the revised manuscript.

Sincerely yours,

Qiyuan Wang
Institute of Earth Environment, Chinese Academy of Sciences
Address: 97 Yanxiang Road, Yanta District, Xi'an 710061, China.
E-mails: wangqy@ieecas.cn
Tel: +86-29-62336205
Fax: +86-29-62336234

**Responses to Referee #1:**

*General comments:*

*Urbanization even anthropogenic activities is an important way to influence air pollution by emissions (gases and particles), meteorological conditions and atmospheric processes (urban heat island), etc. Anthropogenic pollutants include greenhouse gases, gaseous and particulate pollutants. Aerosol is very important to impact atmospheric cycle and climate system by direct and indirect effects, a hot issue of scientific researches internationally. Also, atmospheric pollutions cause adverse harm to human health. Aerosols are known to originate from direct emission and secondary formation, namely, POA and SOA. The organic aerosol (OA) is a very important part of aerosols, including BC and OC. Inorganic ions are important compositions of aerosols. This paper used the data of aerosol optical properties, chemical composition, meteorological parameters used in Xi'an to analyze their temporal variation and compare their difference between the normal period and the COVID-19 in 2019, and to estimate the radiation forcing of aerosols. The topic of this paper is of common interest within the scientific community. Although the manuscript includes some important data, however, the quality is somewhat sufficient in the current state to be directly published.*

**Response:** We thank the reviewer for the careful evaluation of our manuscript. We have revised the manuscript and provided more elaborations on the datasets according to both reviewers. Besides, the language has been also polished by a native English speaker. We believe that the quality of the revised manuscript has been greatly improved.

**Responses to Referee #2:**

*General comments:*

*This work analyzes the impact of the COVID-19 lockdown in China on some atmospheric properties, in particular on the extinction, scattering and absorbing coefficients together with the direct radiative effect, all of them considering the aerosol chemical composition. The topic is clearly in the scope of Atmospheric Chemistry and Physics, and absolutely relevant for the scientific community and decision-makers. The manuscript is very well written, with only few typos. My main concerns (general comments) are three:*

**Response:** We appreciate the thoughtful and valuable suggestions by the reviewer, which are helpful for us to improve the quality of our manuscript. We have addressed the comments in point-by-point form as shown below.

*Comment (1): Title does not reflect the actual content of the work. The current title is quite ambiguous, leading the reader to expect a study on the entire Chinese territory. I suggest to explicit that the analysis focuses on the study case in Xi'an.*

**Response:** Suggestion taken. The title has been revised to "Measurement report: The importance of biomass burning in light extinction and direct radiative effect of urban aerosol during the COVID-19 lockdown in Xi'an, China".

*Comment (2): the sampling campaign consisted of two distinct periods, the so-called normal period (1 to 23 January, 2020) and COVID-19 lockdown period (27 January to 7 February, 2020). Because the aim of the study is to compare the atmosphere during lockdown period against the 'normal' conditions, I consider the normal period chosen here inappropriate. Would it not be more correct to compare with the historical period 27 January 7 February (i.e. average of several years to minimize the effect of different meteorological conditions)?*

**Response:** Thank you for pointing this out. We do agree that a comparison between the COVID-19 lockdown period and the historical same period could minimize the uncertainty of the effects of meteorological conditions. Our intensive online measurement was performed from January 1$^{st}$ to February 9$^{th}$, 2020. Unfortunately, after consulting the local researchers and authorizations and full literature searching on the databases, there was no matching data availability during the same period before 2020. In this study, the generalized additive model analysis that was described in Section 3.2 indicated the reduction of aerosol light extinction from normal period to lockdown period was credited to anthropogenic emission reduction. The normal period of January 1$^{st}$ to 23$^{rd}$, 2020 that did not disturb by any special event and was the closest time theme acted as a reasonable and appropriate reference period to investigate the impacts of anthropogenic emission on the optical property and direct radiative effect of aerosol during the lockdown period.

*Comment (3): there are a lot of figures as supplementary material. Please considered to move some of them to the main manuscript. I suggest to include figures S8, S9 and S13.*

**Response:** Following the reviewer's suggestion, Figures S8 and S9 have been moved to the main manuscript and assigned as Figures 1 and 2, respectively. Figure S13 shows the realistic AAE of BC during the normal and lockdown periods, which was used for the calculation with the Absorption Ångström exponent method (see details in Text S1 as supplementary material). Therefore, we believe that Figure S13 is more appropriate to be kept in the supplementary material as well.

***Specific comments:***
*Comment (4): Line 15: replace 'optical properties of aerosol' by 'aerosol optical properties'.*

**Response:** The phrase "optical properties of aerosol" has been replaced by "aerosol optical properties".

*Comment (5): Lines 39-40: Specify that this sentence refers only to China or include other references with studies worldwide (e.g. Ibrahim et al., 2021). Ibrahim, S., Landa, M., Pešek, O., Pavelka, K., & Halounova, L. (2021). Space-Time Machine Learning Models to Analyze COVID-19 Pandemic Lockdown Effects on Aerosol Optical Depth over Europe. Remote Sensing, 13(15), 3027.*

**Response:** In the revised manuscript, more important worldwide references have been added (e.g., Ibrahim et al., 2021; Kumar et al., 2021; Sanap, 2021; Weber et al., 2020). Besides, our original phrase "recent aerosol studies" refers to China only. The relevant description has been revised as follows:

Page 2 Line 34–44: "The abrupt outbreak of Coronavirus Disease 2019 (COVID-19) caused unprecedented economic and social disruption (Yao et al., 2020). Most worldwide countries implemented the city lockdown to curb the virus spread among humans, providing a rare opportunity to investigate the impacts of anthropogenic activities on the air quality (Ibrahim et al., 2021; Kumar et al., 2021; Sanap, 2021; Weber et al., 2020). The Chinese government also enforced a series of strict restrictions on travel, transport, manufacture, and constructive activities during the lockdown. Recent studies on the aerosols in China which were conducted during the lockdown period focused on primary emissions and secondary formation, and most of them had revealed changes in aerosol compositions, sources, and processes under a variety of emission control measures (Le et al., 2020; Li et al., 2020; Wang et al., 2020a; Wang et al., 2020c; Zhao et al., 2020; Zheng et al., 2020)."


**Response:** Thank you for pointing out these points. For the first point, the relative humidity (RH) is redundant since it can be computed from dew point and air temperature. In the revised manuscript, we have removed RH meteorological parameter and performed the generalized additive model (GAM) analysis again.

For the second point, taking radiation-related meteorological data (e.g., solar radiation, land-surface radiation, net radiation, and etc.) as inputs for the GAM model analysis might improve the accuracy of estimation. Unfortunately, we could not be able to collect the synchronous radiation-related meteorological data during the campaign. The GAM constructed in this study by the existing meteorological parameters (wind speed, wind direction, temperature, pressures, dew point, and planetary boundary layer height) was proved to be reasonable and reliable for estimating aerosol light extinction.

Third, the word "pressures" has been replaced by "pressure". Similar errors in Tables S1, S2, and S3 have been also corrected.

Lastly, to obtain the PBLH at the sampling site (34°13' N, 108°52' E) with the GDAS data, linear interpolation was used. We have added the description of the method in Table S1 as "PBLH at the sampling site was obtained using linear interpolation method."

*Comment (9): Line 181: what is the advantage of using SBDART instead of other shortwave radiative models such as libRadtran?*

**Response:** The Santa Barbara DISORT Atmospheric Radiative Transfer (SBDART) is a software tool for radiative transfer calculations. All the important processes that affect the ultraviolet, visible, and infrared radiation fields are considered. Its code is a marriage of a sophisticated discrete ordinate radiative transfer (DISORT) module, low-resolution transmission (LOWTRAN) models, and the Mie scattering results, which is well suited for a wide variety of atmospheric radiative energy balances and remote sensing studies (Ricchiazzi et al., 1988). The libRadtran is also a widely used software package for various applications related to atmospheric radiation (Emde et al., 2016; Mayer and Kylling, 2005). Its input options have the same DISORT (Stamnes et al., 2000) and LOWTRAN (Pierluissi and Peng, 1985) as the SBDART, and their reliabilities for estimating the irradiance under different aerosol conditions had been validated (Obregón et al., 2015).

Compared to the SBDART adopting Mie theory applied for the equivalent spherical particle, the libRadtran shows the advantage that six individual habits of ice crystal can be used in the model, including plate, solid column, hollow column, rosette 4, rosette 6, and rough aggregate. With the highly reliable-quality ice crystal information, the libRadtran might reduce the uncertainty in calculating the radiative forcing of the cloud. It is well known that the atmospheric aerosol can perturb the Earth's radiative balance indirectly by acting as cloud condensation nuclei; thus, the libRadtran would be preferred when we study the indirect radiation effect of aerosol in the future.


*Comment (10): Line 644: 100 times is missing in this definition of 'change ratio'.*
**Response:** Sorry for the typo, the footnote in Table 1 has been corrected.

**Responses to Community #1:**

*General comments:*

This manuscript, titled by The importance of biomass burning in light extinction and direct radiative effect of urban aerosol during the COVID-19 lockdown in China, investigated the impacts of COVID-19 lockdown on aerosol light extinction and direct radiative effect. In fact, many studies have been conducted to explore the impacts of lockdown measures on the aerosol compositions, but how the lockdown measures influenced the aerosol optical property and direct radiative effect is limited. Though the information given in this study has been well known, it is suitable to publish on the Measurement Report. I suggest a major revision of this paper. Before acceptance, some issues must be clarified.

**Response:** We highly appreciate the thoughtful and valuable suggestions. We have revised the manuscript accordingly, which has been significantly improved. Besides, the language has been also polished by a native English speaker. Please refer to our point-point responses as follow.

*Comment (1): The contributions of biomass burning to aerosol $b_{ext}$ and DRF increased during lockdown period in this study which just focused on only one site. How did you find out that the importance of control biomass burning for tackling climate change in China? More evidence in the other region should be provided to make your conclusion robust.*

**Response:** Our manuscript focused on a study case in Xi'an, China. Therefore, we do agree with the point that it is inappropriate to conclude the importance of controlling biomass burning for tackling climate change throughout the entire of China. Actually, biomass burning is an important anthropogenic source of aerosol in many Chinese cities, such as Tianjin (Khan et al., 2021), Jinan (Cheng et al., 2021), Guangzhou (Huang et al., 2018), Chengdu (Li et al., 2017), and etc. The results and conclusions of this case study could be a significant reference to other cities in China, where the air is greatly polluted by biomass burning. To address this point clear, we made some changes in the revised manuscript as follows:

(1). The title has been revised to specify the location of this case study. It now reads: "Measurement report: The importance of biomass burning in light extinction and direct radiative effect of urban aerosol during the COVID-19 lockdown in Xi'an, China"

(2). To illustrate the importance of controlling biomass burning in China, the sentences have been added in Section 3.5. Page 12 Line 330–335: "Similar to Xi'an city, the pollution sources of traffic and biomass burning were the two most significant anthropogenic sources of aerosol in most Chinese cities, such as Chengdu, Guangzhou, Jinan, Tianjin, and etc (Cheng et al., 2021; Huang et al., 2018; Khan et al., 2021; Li et al., 2017b). The results in this study indicated that the control measures on traffic in the lockdown were highly effective for mitigating the effects of climate change in the short term, while future emission control policies should consider the importance of biomass burning to tackle climate change in China."


[Figure]

**Figure R1.** Linear relationship between the modeled source and the measured PM$_{2.5}$ mass concentration. The modeled source PM$_{2.5}$ was strongly correlated linearly with the measured optical PM$_{2.5}$ ($R^2 = 0.95$, slope $= 0.96$), indicating that the six identified sources can adequately account for the variability in PM$_{2.5}$ mass concentration.

[Figure]

**Figure R2.** Linear relationships between the modeled source and the measured optical (a) $b_{scat}$, (b) $b_{abs}$, and (c) $b_{ext}$. The modeled source $b_{scat}$, $b_{abs}$, and $b_{ext}$ were strongly correlated linearly with the measured optical $b_{scat}$ ($R^2 = 1.00$, slope $= 1.00$), $b_{abs}$ ($R^2 = 0.99$, slope $= 0.99$), and $b_{ext}$ ($R^2 = 1.00$, slope $= 1.00$), indicating that the six identified sources can adequately account for the variability in aerosol optical coefficients.

*Comment (6): Some details such as line 201 "which is" or "which are", should be checked carefully.*
**Response:** Corrected.

[revised manuscript text omitted]
̲s̲, w̲h̲i̲c̲h̶t̶h̶a̶t̶ led to biomass burning bec̲o̲m̲i̲n̲g̶a̶m̶e̶ the most important source to $b_{ext}$ (36.7%) in the lockdown (Figure 4̶6). Hence, additional actions and investigations on biomass burning emissions would be taken into consideration.

**3.5 Impacts of COVID-19 lockdown on aerosol DRE**

Figure 6̶ ̲8̲ shows the range of source-specific aerosol DRE$_{top}$, DRE$_{surface}$, and DRE$_{atmosphere}$ during the normal and lockdown periods. For all sources, the aerosol DRE$_{atmosphere}$ values in both periods were positive, producing net warming effects in the atmosphere. The mean aerosol DRE$_{atmosphere}$ ̲d̲e̲c̲r̲e̲a̲s̲e̲d̶r̶e̶d̶u̶c̶e̶d̶ from 31.0 ±23.2 W m$^{-2}$ before the lockdown to 14.1 ± 11.5 W m$^{-2}$ in the lockdown, with a reduction of 54.5%̲.̲ ̲T̲h̲i̲s̲ ̲c̲a̲n̲ ̲b̲e̲ ̲e̲x̲p̲l̲a̲i̲n̲e̲d̲ ̲b̲y̲ ̲t̲h̲e̲ ̲r̲e̲d̲u̲c̲e̲d̲ ̲a̲e̲r̲o̲s̲o̲l̲ ̲c̲o̲n̲c̲e̲n̲t̲r̲a̲t̲i̲o̲n̲ ̲a̲n̲d̲ ̲i̲n̲c̲r̲e̲a̲s̲e̲d̲ ̲S̲S̲A̲ ̲(̲L̲i̲u̲ ̲e̲t̲ ̲a̲l̲.̲,̲ ̲2̲0̲2̲0̲)̲.

[revised manuscript text omitted]

**Text S1.** Absorption Ångström exponent method

In this study, aerosol light absorption coefficient ($b_{abs}$) at wavelengths of $\lambda$ = 370 nm, 470 nm, 520 nm,

590 nm, 660 nm, and 880 nm were measured by a newly developed Aethalometer (model AE33, Magee

Scientific, Berkeley, CA, USA). The Absorption Ångström exponent (AAE) describes the wavelength dependence of aerosol light absorption, and can be calculated according to power law fitting  $b_{abs}$

at wavelengths of 370 nm to 880 nm (Moosmüller et al., 2011) as below:

$$b_{abs}(\lambda) \sim \lambda^{-AAE} \tag{1}$$

Through the AAE method (Lack and Langridge, 2013), the mass absorption efficiency (MAE) of black carbon (BC) at 520 nm can be obtained as follows:

$$b_{abs}(520 \text{ nm}) = b_{abs\text{-}BC}(520 \text{ nm}) + b_{abs\text{-}BrC}(520 \text{ nm}) \tag{2}$$

$$b_{abs\text{-}BC}(520 \text{ nm}) = b_{abs\text{-}BC}(880 \text{ nm}) \times \left(\frac{520}{880}\right)^{-AAE_{BC}} \tag{3}$$

$$MAE_{BC}(520 \text{ nm}) = \frac{b_{abs\text{-}BC}(520 \text{ nm})}{[BC]} \tag{4}$$

where $b_{abs}$-BC and $b_{abs}$-BrC in Mm$^{-1}$ are the light absorption coefficients caused by BC and brown carbon (BrC), respectively; $AAE_{BC}$ is the AAE caused by the BC particle, which can vary from 0.8 to

1.4 due to core size, coating materials, and mixing state (Lack and Cappa, 2010; Lack and Langridge,

2013). The linear relationship between the AAEs and the mass concentration ratios of organic aerosol (OA) to BC is investigated to find the realistic $AAE_{BC}$ during the normal and lockdown periods (Figure

S12) (Yuan et al., 2016); and [BC] is the mass concentration of BC in µg m$^{-3}$.

**Text S2.** Uncertainty of the element concentration

Considering the element concentration measured by the Xact 625 ambient metals monitor with a 1-hour sampling interval, the uncertainty of the element concentration ($u_e$) inputting into the receptor model was estimated as follows (Norris et al., 2014):

$$u_e = \sqrt{(c_e \times 10\%)^2 + (0.5 \times MDL)^2}, \text{ for } c_e > MDL \tag{5}$$

$$u_e = \frac{5}{6} \times MDL, \text{ for } c_e \leq MDL \tag{6}$$

where $c_e$ is the concentration of the element; 10% is the default analytical relative error (Rai et al., 2020); and MDL represents the method detection limit of  the element.

**Text S3.** Diagnostics of HERM solutions

In this study, factor numbers from two2 to eight8 were selected to run in the HERM software. Each factor solution was performed with completely unconstrained profiles at twenty different seeds to explore the possible sources. Detailed information on how the most interpretable factors were selected is presented below.

As shown in Figure S2, the values of $Q/Q_{exp}$ (> 1) decreased as the factor numbers increased. The large

$Q/Q_{exp}$ values in two2- (21.10 ±0.03) and three3-factor (12.29 ±0.01) solutions indicated too few factors were resolved. In the four4-factor solution (Figure S3), Factor 2 identified as biomass burning was characterized by high explained variations (EV) values of POA (56%), LO-OOA (54%), BC (43%), Cl (55%). Factor 3 was regarded as fugitive dust due to significant EV values of Si (100%), Ca (68%), and

Fe (35%). For the Factor 4 assigned to the secondary source, EV values of $NO_3^-$, $SO_4^{2-}$, $NH_4^+$, and MO-

OOA were larger than 30%. It is noted that Factor 1 was associated with the traffic-like source because

$b_{ext}$ from this source showed a moderate correlation with $NO_x$, a tracer of fresh motor vehicle exhaust emission ($R^2 = 0.58$). However, the high EV values of some specific elements (e.g., As (44%) and Se (31%)) in this factor indicated the possible mixture of other fossil fuel sources (e.g., coal combustion).

When five factors were resolved, except traffic-like source (Factor 1), biomass burning (Factor 2), and fugitive dust (Factor 3), the secondary source was split into nitrate plus SOA (Factor 4) and sulfate plus

SOA (Factor 5) sources (Figure S4). The increase to six6-factor solution (Figure S5) showed well separation of traffic-related emission (Factor 1) and coal combustion (Factor 3). The A stronger correlation between $b_{ext}$ from traffic-related emission and $NO_x$ ($R^2 = 0.72$) was found compared to traffic-like factors resolved in four4— and five5- factor solutions ($R^2 = 0.58$). As shown in Figures S6

and S7, further investigations of unconstrained profile solutions with seven7 and —eight8 factors resulted in factor split. The extra split factors possibly came from biomass burning and coal combustion, mainly due to high EV values of K (26%—33%), or As (21%). Despite $b_{ext}$ from coal combustion factors in seven7- and eight8- factor solutions showed the stronger correlation with As ($R^2 = 0.63–0.68$), Se ($R^2$

$= 0.79–0.86$), and Pb ($R^2 = 0.60–0.67$), the profiles identified coal combustion had no POA contribution.

Meanwhile, the values of POA in fugitive dust profiles identified in seven7— and eight8- factor solutions were higher than 1 (the reference standard of $PM_{2.5}$). It is indicated that these profiles did no2t match the real world.

Therefore,  as the factor solutions described above, six factors were the most interpretable in our study, including traffic-related emission, biomass burning, coal combustion, fugitive dust, nitrate plus SOA source, and sulfate plus SOA source.

[Figure]

 **Figure S1.** The location of the sampling site in Xi'an, China.

[Figure]

**Figure S2.** Values of $Q/Q_{exp}$ for the unconstrained profile solutions with two2 to eight 8 factors at
twenty different seeds.

[Figure]

**Figure S3.** (a) Profiles and (b) time series plots of the resolved source factors in the four-factor solution. The columns in each factor are the profile that display relative relation of absolute values of variables. The red dot represents the explained variation of species for different factors. The corresponding time trends of chemical tracers also are shown.

[Figure]

**Figure S4.** (a) Profiles and (b) time series plots of the resolved source factors in the five5-factor solution. The columns in each factor areis the profile that displaysing the relative relation of absolute values of variables. The red dot represents the explained variation of species for different factors. The corresponding time trends of chemical tracers also are shown.

[Figure]

**Figure S5.** (a) Profiles and (b) time series plots of the resolved source factors in the six6-factor solution.

The columns in each factor areis the profile that displays theing relative relation of absolute values of variables. The red dot represents the explained variation of species for different factors. The corresponding time trends of chemical tracers also are shown.

[Figure]

**Figure S6.** (a) Profiles and (b) time series plots of the resolved source factors in the seven7-factor solution. The columns in each factor areis the profile that displays theing relative relation of absolute values of variables. The red dot represents the explained variation of species for different factors. The corresponding time trends of chemical tracers also are shown.

[Figure]

Chemical and optical variables                        Date (2020)

**Figure S7.** (a) Profiles and (b) time series plots of the resolved source factors in the eight8-factor solution. The columns in each factor areis the profile that displays theing relative relation of absolute values of variables. The red dot represents the explained variation of species for different factors. The corresponding time trends of chemical tracers also are shown.

[Figure]

**Figure S8.** Hourly variations of light scattering ($b_{scat}$), absorption ($b_{abs}$), and extinction ($b_{ext}$) coefficients, single scattering albedo (SSA), and PM$_{2.5}$ mass concentrations in Xi'an during the normal (1 to 23 January) and COVID-19 lockdown (27 January to 7 February) periods.

[Figure]

**Figure S9.** Variations of single scattering albedo (SSA) as a function of (a) secondary inorganic aerosol (SIA = NH$_4$NO$_3$ + (NH$_4$)$_2$SO$_4$)/PM$_{2.5}$, (b) secondary organic aerosol (SOA = LO-OOA + MO-OOA)/OA, and (c) LO-OOA/MO-OOA ratios during the normal and COVID-19 lockdown periods.

[Figure]

**Figure S8.** Linear relationships between the reconstructed chemical and the measured optical (a) $b_{scat}$, (b) $b_{abs}$, and (c) $b_{ext}$.

[Figure]

**Figure S9.** Linear relationships between $PM_{2.5}$ and PM used in  reconstruction of aerosol optical coefficients. PM is the sum of $NH_4NO_3$, $(NH_4)_2SO_4$, fine soil, BC, POA, LO-OOA, and MO-OOA in this study. The slope of  linear regression between $PM_{2.5}$ and PM concentrations (0.79) was close to that between the measured optical $b_{ext}$ and the reconstructed chemical $b_{ext}$ (0.78, see Figure S8c), suggesting that chemical calculation of $b_{ext}$ was a reasonable estimation of aerosol optical coefficients by using chemical components data.

[Figure]

Figure S10. Linear relationship between the modeled source and the measured $PM_{2.5}$ mass concentration. The modeled source $PM_{2.5}$ was strongly correlated linearly with the measured optical $PM_{2.5}$ ($R^2 = 0.95$, slope = 0.96), indicating that the six identified sources can adequately account for the variability in $PM_{2.5}$ mass concentration.

[Figure]

**Figure S11̶2̶.** Linear relationships between the modeled source and the measured optical (a) $b_{scat}$, (b)

$b_{abs}$, and (c) $b_{ext}$. The modeled source $b_{scat}$, $b_{abs}$, and $b_{ext}$ w̲e̲r̲e̲a̶s̶ strongly correlated linearly with the measured optical $b_{scat}$ ($R^2 = 1.00$, slope = 1.00), $b_{abs}$ ($R^2 = 0.99$, slope = 0.99), and $b_{ext}$ ($R^2 = 1.00$, slope

= 1.00), indicating that the six identified sources can adequately account for the variability in aerosol optical coefficients̲$b̶_{ext}̶$.

[Figure]

[Figure]

[Figure]

**Figure S13.** Linear relationships between the AAEs and the mass concentration ratios of organic aerosol (OA) to BC (OA/BC) during the normal (a) and lockdown (b) periods. The intercept of the linear regression represents the realistic $AAE_{BC}$. The points and light gray shadows represent the mean values and error margins in each bin ($\Delta(OA/BC) = 0.5$).

**Table S1.** Summary of chemical and meteorological measurements of in Xi'an before and during the

COVID-19 lockdown period.

| Parameters | Sampling interval | Instruments and online source | Operation and calibration |
|---|---|---|---|
| **Chemical variables** | | | |
| $NO_3^-$, $SO_4^{2-}$, $NH_4^+$, $Cl^-$, and OA | 15-min | Quadrupole aerosol chemical speciation monitor (Q-ACSM, Aerodyne Research Inc., Billerica, Massachusetts, USA) | The relative ionization efficiencies (RIEs) for OA, nitrate, and chloride were set to 1.4, 1.1, and 1.3 by default, respectively. The RIE for ammonium (5.8) was determined from the ammonium nitrate aerosol calibration, while the RIE for sulfate (1.9) was estimated by fitting the measured sulfate versus predicted sulfate values. The collection efficiency was set to 0.45. |
| Si, K, Ca, Cr, Mn, Fe, Zn, As, Se, Ba, Hg, and Pb | 1-hour | Xact 625 ambient metals monitor (Xact 625i, Cooper Environmental Services, Beaverton, OR, USA) | Daily advanced quality assurance checks were performed during 30 min after midnight to monitor shifts in the calibration. |
| $PM_{2.5}$ and $NO_x$ | 5-min | The Department of Ecology and Environment of Shaanxi Province (http://sthjt.shaanxi.gov.cn, in Chinese) | |
| **Meteorological variables**[*] | | | |
| WS, WD, , T, P, and DP | 1-hour | Integrated automatic weather station (MAWS201, Vaisala, Helsinki, Finland) | |
| PBLH | 3-hour | Global Data Assimilation System (ftp://arlftp.arlhq.noaa.gov/pub/archives/gdas1) | PBLH at the sampling site was obtained using linear interpolation method. |

[*]WS, WD, , T, P, DP, and PBLH represent wind speed, wind direction, temperature, pressure, dew point, and planetary boundary layer height, respectively.

**Table S2.** Summary of output indices from the constructed $b_{ext}$ GAM.

| | | |
|---|---|---|
| Intercept | 6.64 | |
| Adjusted $R^2$ | 0.54 | |
|  |  | |

| Smoothed parameters[*] | F value | $p$ value |
|---|---|---|
| $f$(WS) | 3.402 | 0.002331 |
| $f$(WD) | 5.820 | 0.000134 |
|  |  |  |
| $f$(T) | 2.707 | 0.01280 |
| $f$(P) | 3.209 | 0.001757 |
| $f$(DP) | 13.325 | $\leq 2.00\times10^{-16}$ |
| $f$(PBLH) | 3.656 | 0.026822 |

[*]WS, WD,  T, P, DP, and PBLH represent wind speed, wind direction, temperature, pressure, dew point, and planetary boundary layer height, respectively.

**Table S3.** Concurvity indices between each independent smoothed parameter in the constructed GAM.

| Smoothed parameters[*] | f(WS) | f(WD) | f(T) | f(P) | f(DP) | f(PBLH) |
|---|---|---|---|---|---|---|
| f(WS) | 1.00 | 0.28 | 0.03 | 0.09 | 0.07 | 0.23 |
| f(WD) | 0.15 | 1.00 | 0.08 | 0.09 | 0.03 | 0.07 |
| f(T) | 0.06 | 0.07 | 1.00 | 0.11 | 0.25 | 0.22 |
| f(P) | 0.08 | 0.24 | 0.08 | 1.00 | 0.06 | 0.09 |
| f(DP) | 0.05 | 0.06 | 0.08 | 0.07 | 1.00 | 0.05 |
| f(PBLH) | 0.13 | 0.07 | 0.05 | 0.04 | 0.06 | 1.00 |

| Smoothed parameters[*] | f(WS) | f(WD) | f(RH) | f(T) | f(P) | f(DP) | f(PBLH) |
|---|---|---|---|---|---|---|---|
| f(WS) | 1.00 | 0.27 | 0.09 | 0.03 | 0.09 | 0.08 | 0.23 |
| f(WD) | 0.16 | 1.00 | 0.05 | 0.08 | 0.10 | 0.03 | 0.07 |
| f(RH) | 0.08 | 0.10 | 1.00 | 0.04 | 0.03 | 0.36 | 0.33 |
| f(T) | 0.08 | 0.07 | 0.14 | 1.00 | 0.12 | 0.23 | 0.22 |
| f(P) | 0.09 | 0.23 | 0.07 | 0.05 | 1.00 | 0.07 | 0.09 |
| f(DP) | 0.05 | 0.06 | 0.36 | 0.08 | 0.06 | 1.00 | 0.05 |
| f(PBLH) | 0.13 | 0.07 | 0.32 | 0.06 | 0.05 | 0.08 | 1.00 |

[*]WS, WD, RH, T, P, DP, and PBLH represent wind speed, wind direction, relative humidity,
temperature, pressures, dew point, and planetary boundary layer height, respectively.

---

## Referee Report (RR1)

Comments to " The importance of biomass burning in light extinction and direct radiative effect of urban aerosol during the COVID-19 lockdown in Xi'an, China" by Tian et al.

The paper by Tian et al. investigated the impacts of anthropogenic sources on $b_{ext}$ and direct radiative forcing (DRF). They found out that biomass burning dominated $b_{ext}$ and DRF during the COVID-19 lockdown in Xi'an, China. This paper is well-written and topical. However, some important details on the measurement methods and data analysis method are needed. Furthermore, some results require further interpretation. I suggest that this paper will be published in ACP after addressing the points listed below.

1. The mass concentrations of ions, OA were measured in $PM_1$. In fact, considerable fractions of them might be distributed in $PM_{1-2.5}$, especially during the polluted period in north China. Thus, ions and OA in $PM_{2.5}$ should be underestimated. I suggest author reconstruct the $PM_{2.5}$ mass based on these measured chemical compositions to discuss their uncertainties.

2. The uncertainties of estimated MSEs and MAEs of chemical compositions in table 2 should be large due to the comment 1.

3. Generally, the formation mechanisms of $NH_4NO_3$ and $(NH_4)_2SO_4$ might be related with aqueous chemical processes. In Mie theory, their MSEs might be similar due to their similar size distributions. However, why their estimated MSEs are very different especially during the normal period?

4. I suggest the author to further analyze the possible source or formation mechanisms of LO-OOA and MO-OOA. At least it needs to be discussed whether these OOAs came from the oxidation of POA or directly from the oxidation of VOCs. According to the estimated MSEs of POA, LO-OOA and MO-OOA, mass median diameter of POA might be evidently lower than those of LO-OOA and MO-OOA. Why the MSEs of LO-OOA and MO-OOA is higher than that of POA needs further discussion. In addition, why the MSE of MO-OOA during the lockdown period is higher than that during the normal period also needs further discussion.

5. The combined contribution of POA, LO-OOA and MO-OOA to $b_{ext}$ was over 60% and the combined contribution of $NH_4NO_3$ and $(NH_4)_2SO_4$ was over 20% during the during the lockdown period in Fig.4. In contrast, the contribution of biomass burning to $b_{ext}$ was only 37% during the lockdown period in Fig.6. This means that secondary organics and secondary inorganics from gaseous precursors (e.g., $SO_2$, NOx and VOCs) emitted from coal combustion contributed slightly more to $b_{ext}$ than biomass burning. Therefore, controlling biomass burning is as important as coal combustion in this city.

6. The contributions of six sources to DRE were estimated under the dry condition. To some extent, the contribution of coal combustion to $b_{ext}$ might be significantly higher than that of biomass burning under ambient RH condition due to the hygroscopic growth according to comment 5.

---

## Referee Report (RR2)

Comments to " The importance of biomass burning in light extinction and direct radiative effect of urban aerosol during the COVID-19 lockdown in Xi'an, China" by Tian et al. First, the authors have done their best to respond my comments and I have no further comments. Second, I expect the authors to address unanswered comments in future work. Finally, I suggest that this paper cloud be published in ACP in this revised version.

---

## Author Response (AR2)

**Responses to the Editor:**

Comments to the author: There are major concerns by one of the referees that should be addressed so please consider these suggestion/clarification for a further review process.

Non-public comments to the Author: There are major concerns by one of the referees that should be addressed so please consider these suggestion/clarification for a further review process.

**Response:** Thank you for providing us the opportunity to revise and improve our manuscript. We are also grateful for the valuable comments and suggestions by the reviewers. We have addressed all the raised concerns with care. Below are point-to-point responses. Detailed responses to each of the reviewer's comments are provided in blue, and the revised text is underlined. Attached please also find the marked-up manuscript to track the changes in the revised manuscript.

**Responses to Referee #3:**

No suggestion for revision.

**Response:** We thank the reviewer for taking the time to assess this manuscript.

**Responses to Referee #4:**

**General comments:**

Comments to "The importance of biomass burning in light extinction and direct radiative effect of urban aerosol during the COVID-19 lockdown in Xi'an, China" by Tian et al.

The paper by Tian et al. investigated the impacts of anthropogenic sources on $b_{ext}$ and direct radiative forcing (DRF). They found out that biomass burning dominated $b_{ext}$ and DRF during the COVID-19 lockdown in Xi'an, China. This paper is well-written and topical. However, some important details on the measurement methods and data analysis method are needed. Furthermore, some results require further interpretation. I suggest that this paper will be published in ACP after addressing the points listed below.

**Response:** We thank the reviewer for the helpful comments and providing us the opportunity to strengthen our research. For the concerns regarding the measurement methods and data analysis method, we have carefully addressed them in the responses of comments (1) and (2). Furthermore, we improved the results and discussion following the reviewer's comments. We hope the revised manuscript facilitates the understanding of the readers. Please see our point-to-point responses as follow.

**Comment (1):** The mass concentrations of ions, OA were measured in $PM_1$. In fact, considerable fractions of them might be distributed in $PM_{1-2.5}$, especially during the polluted period in north China. Thus, ions and OA in $PM_{2.5}$ should be underestimated.

I suggest author reconstruct the PM$_{2.5}$ mass based on these measured chemical compositions to discuss their uncertainties.

**Response:** We agree with the reviewer that OA and ions concentrations measured in PM$_1$ might be lower than those in PM$_{2.5}$. We thus have compared the measured PM$_{2.5}$ mass concentration with the sum concentration of POA, LO-OOA, MO-OOA, NH$_4$NO$_3$, (NH$_4$)$_2$SO$_4$, BC and fine soil (the sum is referred to as the reconstructed PM), following the reviewer's suggestion. We find the measured PM$_{2.5}$ and the reconstructed PM are strongly correlated with an R$^2$ of 0.86, however, the slope of this linear regression is only 0.79 (Figure S2). That is, the mass concentration of reconstructed PM is 0.79 times of the measured PM$_{2.5}$. In the revised manuscript, we have added the above discussion in the method Sect. 2.4:

> "The reconstructed [PM] ([PM] = [NH$_4$NO$_3$] + [(NH$_4$)$_2$SO$_4$] + [POA] + [LO-OOA] + [MO-OOA] + [BC] + [fine soil]) is strongly correlated with the measured [PM$_{2.5}$] (R$^2$=0.86), with a slope of 0.79 (Figure S2). That is, the reconstructed [PM] accounted for ~79% of the measured [PM$_{2.5}$]." (*Page 6 Line 155–157*)

In the revised supplement, Figure S2 shows:

[Figure]

**Figure S2**. Linear relationship between the measured PM$_{2.5}$ concentration and the sum concentration of POA, LO-OOA, MO-OOA, NH$_4$NO$_3$, (NH$_4$)$_2$SO$_4$, BC and fine soil (the sum is referred to as the reconstructed PM).

The consequence of unaccounted mass of OA and ions in PM$_{1-2.5}$ on the calculated MSEs and MAEs are discussed and addressed in the comment (2).

**Comment (2):** The uncertainties of estimated MSEs and MAEs of chemical compositions in table 2 should be large due to the comment 1.

**Response:** Thank you for pointing this point. In this study, we measured OA and ions in PM$_1$ using Q-ACSM (method Sect. 2.2.2). However, a fraction of OA and ions can be distributed in PM$_{1-2.5}$, as the reviewer pointed out in the comment (1), which is not considered in our analysis and thus will affect the calculation of the MSEs and MAEs following equations (2) and (3).

To address this concern, we conduct a sensitivity analysis with respect to the OA and ions mass concentrations. Their mass fraction in $PM_{1-2.5}$ is not known, and we use the fractional difference between the measured $PM_{2.5}$ and the reconstructed PM mass to represent it (i.e., $1-0.79=0.21$). This value is in consistent with previous study in China (Sun et al., 2020). The analysis shows that if the OA and ions mass increased by 21% to include their mass in $PM_{1-2.5}$, the estimated MSEs and MAEs will correspondingly decrease by 21%. The moderate changes in mass concentrations of OA and ions will not significantly change any conclusions made in this study, because (1) both MSEs of $NH_4NO_3$, $(NH_4)_2SO_4$, POA, LO-OOA, MO-OOA, and MAEs of POA, LO-OOA, MO-OOA in Table 2 are overestimated by the similar extent, thus will not affect the comparison of MSEs and MAEs in Sect. 3.3; (2) the $b_{ext}$ ($=b_{scat}+b_{abs}$) of chemical species, the product of the mass concentration and (MSEs+MAEs), does not change in the sensitivity analysis, compared with the results in Figure 4. The above sensitivity analysis has little effect on the MSEs of fine soil, because the fine soil is in $PM_{2.5}$ in this study (Sect. 2.2.2). The MAEs of BC is not affect, because it is also in $PM_{2.5}$ (Sect. 2.2.1); and is calculated by the independent absorption Angstrom exponent method (Sect. 2.4; Text S1).

In the revised manuscript, we add the following discussion in the main text:

> "The $NH_4NO_3$, $(NH_4)_2SO_4$, POA, LO-OOA, and MO-OOA in $PM_{1-2.5}$ are not included in the calculation (equations 2 and 3). A sensitivity analysis concludes that if their concentrations increased by 21% to match the measured $PM_{2.5}$, then the estimated MSEs and MAEs will correspondingly decrease by 21%." (*Page 6 Line 157–160*)

**Comment (3):** Generally, the formation mechanisms of $NH_4NO_3$ and $(NH_4)_2SO_4$ might be related with aqueous chemical processes. In Mie theory, their MSEs might be similar due to their similar size distributions. However, why their estimated MSEs are very different especially during the normal period?

**Response:** MSEs of $NH_4NO_3$ and $(NH_4)_2SO_4$ would be similar if they have similar particle diameters in Mie theory, considering that they have similar refraction indices and particle densities. However, we did not measure the particle size in this study campaign. Previous studies found different size distributions of $NO_3^-$ and $SO_4^{2-}$ measured in $PM_1$ in urban China in winter (Hu et al., 2017; Zhang et al., 2013; Zhu et al., 2021). Our previous study on the same campaign found that $(NH_4)_2SO_4$ was related with aqueous chemical processes, but $NH_4NO_3$ was also related with both aqueous chemical processes and photochemical oxidation (Tian et al., 2021). The different formation pathway of $(NH_4)_2SO_4$ and $NH_4NO_3$ could lead to their different size distributions. Therefore, the different MSEs of $NH_4NO_3$ and $(NH_4)_2SO_4$ were probably caused by their different size distributions. The differences in MSEs between $NH_4NO_3$ and $(NH_4)_2SO_4$ are larger during the normal period than those during the lockdown period. However, we could not over-interpret this due to the lack of measured particle sizes.

To make this point clear, we add the following discussion in the revised manuscript:

"NH$_4$NO$_3$ and (NH$_4$)$_2$SO$_4$ had different MSEs, probably due to their different size distributions (Hu et al., 2017; Zhang et al., 2013b; Zhu et al., 2021)." (*Page 9 Line 241–242*)

In the reference list, we add:

"Hu, W., Hu, M., Hu, W. W., Zheng, J., Chen, C., Wu, Y. S., and Guo, S.: Seasonal variations in high time-resolved chemical compositions, sources, and evolution of atmospheric submicron aerosols in the megacity Beijing, Atmos. Chem. Phys., 17, 9979–10000, https://doi.org/10.5194/acp-17-9979-2017, 2017.

Zhang, Y. M., Sun, J. Y., Zhang, X. Y., Shen, X. J., Wang, T. T., and Qin, M. K.: Seasonal characterization of components and size distributions for submicron aerosols in Beijing, Sci. China-Earth Sci., 56, 890–900, https://doi.org/10.1007/s11430-012-4515-z, 2013b.

Zhu, W. F., Zhou, M., Cheng, Z., Yan, N. Q., Huang, C., Qiao, L. P., Wang, H. L., Liu, Y, C., Lou, S. R., and Guo, S.: Seasonal variation of aerosol compositions in Shanghai, China: Insights from particle aerosol mass spectrometer observations, Sci. Total Environ., 771, 144948, https://doi.org/10.1016/j.scitotenv.2021.144948, 2021."

**Comment (4): (a)** I suggest the author to further analyze the possible source or formation mechanisms of LO-OOA and MO-OOA. At least it needs to be discussed whether these OOAs came from the oxidation of POA or directly from the oxidation of VOCs. **(b)** According to the estimated MSEs of POA, LO-OOA and MO-OOA, mass median diameter of POA might be evidently lower than those of LO-OOA and MO-OOA. Why the MSEs of LO-OOA and MO-OOA is higher than that of POA needs further discussion. **(c)** In addition, why the MSE of MO-OOA during the lockdown period is higher than that during the normal period also needs further discussion.

**Response: (a)** In our previous study on the same campaign, we found that LO-OOA/ΔCO ratio was correlated well with both O$_x$ (NO$_2$ + O$_3$) and aerosol liquid water content (ALWC). Same correlations were also found for MO-OOA. This indicates that both photochemical oxidation and aqueous-phase reaction were important pathways for the formation of LO-OOA and MO-OOA (Tian et al., 2021).

As the reviewer's comments, both the oxidation of POA and of VOCs contribute to the OOAs. In this study, OOAs are resolved using receptor models with mass-to-charge (*m/z*) fragments from 12 to 120 derived from measured OA data (method Sect. 2.2.2), identified by the mass spectra with high signal at *m/z* 44. Using this method, it is challenging in distinguishing the contribution of POA and VOCs to OOAs formation (Canonaco et al., 2013).

**(b)** Previous studies found increased mass diameter of OA with increasing OA oxidation level (Wang et al., 2021; Xu et al., 2015). We agree with the reviewer that the MSEs increase from POA to MO-OOA could be attributed to their particle size growth. In the revised manuscript, to make it clear, we add:

> "The MSEs of OA increased with oxidation level from POA to MO-OOA. This could be explained by the increased mass diameter of OA with increasing OA oxidation level (Wang et al., 2021; Xu et al., 2015)." *(Page 9 Line 248–249)*

**(c)** In this study, the MSE of MO-OOA during the lockdown period was larger than that during the normal period. This is probably related to the enhanced atmospheric oxidation capacity during the lockdown period of this sampling campaign (Tian et al., 2021), which is also found in previous observations (Huang et al., 2020; Le et al., 2020). This results in the higher aging state of OA with larger diameter (Hu et al., 2016; Zhu et al., 2021), thus larger MSE during the lockdown period. We add this explanation to the revised Sect. 3.3:

> "The MSE of MO-OOA was larger during the lockdown period than that during the normal period, probably related to the enhanced atmospheric oxidation capacity during the lockdown period (Tian et al., 2021), which is found to result in higher aging state of OA with larger diameter (Hu et al., 2016; Zhu et al., 2021)." *(Page 9 Line 249–252)*

In the reference list, we add the new reference:

> "Hu, W. W., Hu, M., Hu, W., Jimenez, J. L., Yuan, B., Chen, W. T., Wang, M., Wu, Y. S., Chen, C., Wang, Z. B., Peng, J. F., Zeng, L. M., and Shao, M.: Chemical composition, sources, and aging process of submicron aerosols in Beijing: Contrast between summer and winter, J. Geophys. Res.-Atmos., 121, 1955–1977, https://doi.org/10.1002/2015JD024020, 2016."

**Comment (5):** The combined contribution of POA, LO-OOA and MO-OOA to $b_{ext}$ was over 60% and the combined contribution of $NH_4NO_3$ and $(NH_4)_2SO_4$ was over 20% during the lockdown period in Fig. 4. In contrast, the contribution of biomass burning to $b_{ext}$ was only 37% during the lockdown period in Fig. 6. This means that secondary organics and secondary inorganics from gaseous precursors (e.g., $SO_2$, $NO_x$ and VOCs) emitted from coal combustion contributed slightly more to $b_{ext}$ than biomass burning. Therefore, controlling biomass burning is as important as coal combustion in this city.

**Response:** We would like to stress that gaseous precursors (e.g., $SO_2$, $NO_x$ and VOCs) are not solely from coal combustion, but also from biomass burning and vehicular emissions (Wang et al., 2017; Zhou et al., 2017). To our best knowledge, the receptor model method cannot quantify how much secondary organic and inorganic are from gaseous precursors emitted from coal combustion. As shown in Figure 6, during the lockdown period, the contribution of secondary sources to $b_{ext}$ (i.e., sulfate plus SOA source, 37.9%; nitrate plus SOA source, 11.8%) is larger than that of biomass burning (36.7%); however, we could not further attribute secondary sources to their respective primary source, e.g., coal combustion, biomass burning or vehicle emissions.

In this study, we make a conclusion that biomass burning is important to $b_{ext}$ during the lockdown period, but not coal combustion, because (a) the contribution of biomass burning to $b_{ext}$ (36.7%) is larger than coal combustion (6.7%), (b) the $b_{ext}$ from biomass burning did not change significantly from normal to lockdown period, whereas the $b_{ext}$ from coal combustion decreased by a factor of 2 from $73.6 \pm 60.9$ Mm$^{-1}$ to $38.5 \pm 34.5$ Mm$^{-1}$. These changes are consistent with the control measures during the lockdown period, as stated in Sect. 3.4.

**Comment (6):** The contributions of six sources to DRE were estimated under the dry condition. To some extent, the contribution of coal combustion to $b_{ext}$ might be significantly higher than that of biomass burning under ambient RH condition due to the hygroscopic growth according to comment 5.

**Response:** Yes, the contributions of six sources to DRE were estimated under the dry condition in this study, because atmospheric particles first pass through a Nafion® dryer before they were sampled by instruments (method Sect. 2.2.1).

As addressed in the comment (5), during the lockdown period the contribution of biomass burning to $b_{ext}$ (36.7%) is much larger than that of coal combustion (6.7%). As shown in Figure 6, the $b_{ext}$ from biomass burning was $215.4 \pm 163.9$ Mm$^{-1}$, 5 times higher than that from coal combustion ($38.5 \pm 34.5$ Mm$^{-1}$) under dry condition. We agree with the reviewer that the $b_{ext}$ under dry condition can be differ from the $b_{ext}$ under ambient RH condition. To estimate the $b_{ext}$ under ambient RH condition, we need to multiply the $b_{ext}$ under dry condition by the $f$(RH) (i.e., hygroscopic growth factor). Under ambient RH condition, if coal combustion has higher $b_{ext}$ than biomass burning, then the $f$(RH) of coal combustion should be at least 5 times higher than that of biomass burning, which is not very likely (Pitchford et al., 2007).

In this study, we focus on the contributions of different sources to $b_{ext}$ and DRE. The influence of $f$(RH) on $b_{ext}$ is beyond the scope of this manuscript. If the reviewer allows, we would prefer not to discuss it, to avoid distracting the readers, if possible. Because we tried to say that the contribution of biomass burning to $b_{ext}$ and DRE is important during the lockdown period, and the influence of $f$(RH) on $b_{ext}$ is not likely to change this conclusion.


$$b_{abs}(\lambda) \sim \lambda^{-AAE} \tag{1}$$

Through the AAE method (Lack and Langridge, 2013), the mass absorption efficiency (MAE) of black carbon (BC) at 520 nm can be obtained as follows:

$$b_{abs}(520 \text{ nm}) = b_{abs\text{-}BC}(520 \text{ nm}) + b_{abs\text{-}BrC}(520 \text{ nm}) \tag{2}$$

$$b_{abs\text{-}BC}(520 \text{ nm}) = b_{abs\text{-}BC}(880 \text{ nm}) \times \left(\frac{520}{880}\right)^{-AAE_{BC}} \tag{3}$$

$$MAE_{BC}(520 \text{ nm}) = \frac{b_{abs\text{-}BC}(520 \text{ nm})}{[BC]} \tag{4}$$

where $b_{abs}$-BC and $b_{abs}$-BrC in Mm$^{-1}$ are the light absorption coefficients caused by BC and brown carbon (BrC), respectively; $AAE_{BC}$ is the AAE caused by the BC particle, which can vary from 0.8 to

1.4 due to core size, coating materials, and mixing state (Lack and Cappa, 2010; Lack and Langridge,

2013). The linear relationship between the AAEs and the mass concentration ratios of organic aerosol (OA) to BC is investigated to find the realistic $AAE_{BC}$ during the normal and lockdown periods (Figure

S12) (Yuan et al., 2016); and [BC] is the mass concentration of BC in μg m$^{-3}$.

**Text S2.** Uncertainty of the element concentration

Considering the element concentration measured by the Xact 625 ambient metals monitor with a 1-hour sampling interval, the uncertainty of the element concentration ($u_e$) inputting into the receptor model was estimated as follows (Norris et al., 2014):

$$u_e = \sqrt{(c_e \times 10\%)^2 + (0.5 \times MDL)^2}, \text{ for } c_e > MDL \tag{5}$$

$$u_e = \frac{5}{6} \times MDL, \text{ for } c_e \leq MDL \tag{6}$$

where $c_e$ is the concentration of the element; 10% is the default analytical relative error (Rai et al., 2020); and MDL represents the method detection limit of the element.

**Text S3.** Diagnostics of HERM solutions

In this study, factor numbers from two to eight were selected to run in the HERM software. Each factor solution was performed with completely unconstrained profiles at twenty different seeds to explore the possible sources. Detailed information on how the most interpretable factors were selected is presented below.

As shown in Figure S2, the values of $Q/Q_{exp}$ (> 1) decreased as the factor numbers increased. The large

$Q/Q_{exp}$ values in two- (21.10 ±0.03) and three-factor (12.29 ±0.01) solutions indicated too few factors were resolved. In the four-factor solution (Figure S3), Factor 2 identified as biomass burning was characterized by high explained variations (EV) values of POA (56%), LO-OOA (54%), BC (43%), Cl (55%). Factor 3 was regarded as fugitive dust due to significant EV values of Si (100%), Ca (68%), and

Fe (35%). For the Factor 4 assigned to the secondary source, EV values of $NO_3^-$, $SO_4^{2-}$, $NH_4^+$, and MO-

OOA were larger than 30%. It is noted that Factor 1 was associated with the traffic-like source because

$b_{ext}$ from this source showed a moderate correlation with $NO_x$, a tracer of fresh motor vehicle exhaust emission ($R^2 = 0.58$). However, the high EV values of some specific elements (e.g., As (44%) and Se (31%)) in this factor indicated the possible mixture of other fossil fuel sources (e.g., coal combustion).

When five factors were resolved, except traffic-like source (Factor 1), biomass burning (Factor 2), and fugitive dust (Factor 3), the secondary source was split into nitrate plus SOA (Factor 4) and sulfate plus

SOA (Factor 5) sources (Figure S4). The increase to six-factor solution (Figure S5) showed well separation of traffic-related emission (Factor 1) and coal combustion (Factor 3). A stronger correlation between $b_{ext}$ from traffic-related emission and $NO_x$ ($R^2 = 0.72$) was found compared to traffic-like factors resolved in four– and five- factor solutions ($R^2 = 0.58$). As shown in Figures S6 and S7, further investigations of unconstrained profile solutions with seven and eight factors resulted in factor split. The extra split factors possibly came from biomass burning and coal combustion, mainly due to high EV

values of K (26–33%), or As (21%). Despite $b_{ext}$ from coal combustion factors in seven- and eight- factor solutions showed the stronger correlation with As ($R^2 = 0.63–0.68$), Se ($R^2 = 0.79–0.86$), and Pb ($R^2 = 0.60–0.67$), the profiles identified coal combustion had no POA contribution. Meanwhile, the values of POA in fugitive dust profiles identified in seven– and eight- factor solutions were higher than

1 (the reference standard of $PM_{2.5}$). It is indicated that these profiles did not match the real world.

Therefore, as the factor solutions described above, six factors were the most interpretable in our study, including traffic-related emission, biomass burning, coal combustion, fugitive dust, nitrate plus SOA

source, and sulfate plus SOA source.

[Figure]

**Figure S1.** The location of the sampling site in Xi'an, China.

[Figure]

**Figure S2.** Linear relationship between the measured PM$_{2.5}$ concentration and the sum concentration of

POA, LO-OOA, MO-OOA, NH$_4$NO$_3$, (NH$_4$)$_2$SO$_4$, BC, and fine soil (the sum is referred to as the reconstructed PM).

[Figure]

Number of factors

**Figure S2S3.** Values of $Q/Q_{exp}$ for the unconstrained profile solutions with two to eight factors at twenty
different seeds.

[Figure]

**Figure S4.** (a) Profiles and (b) time series plots of the resolved source factors in the four-factor solution. The columns in each factor are the profile that displays the relative relation of absolute values of variables. The red dot represents the explained variation of species for different factors. The corresponding time trends of chemical tracers also are shown.

[Figure]

**Figure S5.** (a) Profiles and (b) time series plots of the resolved source factors in the five-factor solution. The columns in each factor are the profile that displays the relative relation of absolute values of variables. The red dot represents the explained variation of species for different factors. The corresponding time trends of chemical tracers also are shown.

[Figure]

**Figure S5S6.** (a) Profiles and (b) time series plots of the resolved source factors in the six-factor solution.

The columns in each factor are the profile that displays the relative relation of absolute values of variables. The red dot represents the explained variation of species for different factors. The corresponding time trends of chemical tracers also are shown.

[Figure]

**Figure S6S7.** (a) Profiles and (b) time series plots of the resolved source factors in the seven-factor solution. The columns in each factor are the profile that displays the relative relation of absolute values of variables. The red dot represents the explained variation of species for different factors. The corresponding time trends of chemical tracers also are shown.

[Figure]

**Figure S8.** (a) Profiles and (b) time series plots of the resolved source factors in the eight-factor solution. The columns in each factor are the profile that displays the relative relation of absolute values of variables. The red dot represents the explained variation of species for different factors. The corresponding time trends of chemical tracers also are shown.

[Figure]

**Figure S98**. Linear relationships between the reconstructed chemical and the measured optical (a) $b_{scat}$, (b) $b_{abs}$, and (c) $b_{ext}$.

[Figure]

**Figure S9.** $_{2.5}$

$_4$$_3$$_4$$_2$$_4$

$_{2.5}$

[Figure]

**Figure S10.** Linear relationship between the modeled source and the measured PM$_{2.5}$ mass
concentration. The modeled source PM$_{2.5}$ was strongly correlated linearly with the measured optical
PM$_{2.5}$ (R$^2$ = 0.95, slope = 0.96), indicating that the six identified sources can adequately account for the
variability in PM$_{2.5}$ mass concentration.

[Figure]

**Figure S11.** Linear relationships between the modeled source and the measured optical (a) $b_{scat}$, (b) $b_{abs}$, and (c) $b_{ext}$. The modeled source $b_{scat}$, $b_{abs}$, and $b_{ext}$ were strongly correlated linearly with the measured optical $b_{scat}$ ($R^2$ = 1.00, slope = 1.00), $b_{abs}$ ($R^2$ = 0.99, slope = 0.99), and $b_{ext}$ ($R^2$ = 1.00, slope = 1.00), indicating that the six identified sources can adequately account for the variability in aerosol optical coefficients.

[Figure]

**Figure S12.** Linear relationships between the AAEs and the mass concentration ratios of organic aerosol (OA) to BC (OA/BC) during the normal (a) and lockdown (b) periods. The intercept of the linear regression represents the realistic $AAE_{BC}$. The points and light gray shadows represent the mean values and error margins in each bin ($\Delta$(OA/BC) = 0.5).

**Table S1.** Summary of chemical and meteorological measurements of in Xi'an before and during the
COVID-19 lockdown period.

| Parameters | Sampling interval | Instruments and online source | Operation and calibration |
|---|---|---|---|
| **Chemical variables** | | | |
| $NO_3^-$, $SO_4^{2-}$, $NH_4^+$, $Cl^-$, and OA | 15-min | Quadrupole aerosol chemical speciation monitor (Q-ACSM, Aerodyne Research Inc., Billerica, Massachusetts, USA) | The relative ionization efficiencies (RIEs) for OA, nitrate, and chloride were set to 1.4, 1.1, and 1.3 by default, respectively. The RIE for ammonium (5.8) was determined from the ammonium nitrate aerosol calibration, while the RIE for sulfate (1.9) was estimated by fitting the measured sulfate versus predicted sulfate values. The collection efficiency was set to 0.45. |
| Si, K, Ca, Cr, Mn, Fe, Zn, As, Se, Ba, Hg, and Pb | 1-hour | Xact 625 ambient metals monitor (Xact 625i, Cooper Environmental Services, Beaverton, OR, USA) | Daily advanced quality assurance checks were performed during 30 min after midnight to monitor shifts in the calibration. |
| $PM_{2.5}$ and $NO_x$ | 5-min | The Department of Ecology and Environment of Shaanxi Province (http://sthjt.shaanxi.gov.cn, in Chinese) | |
| **Meteorological variables**[*] | | | |
| WS, WD, T, P, and DP | 1-hour | Integrated automatic weather station (MAWS201, Vaisala, Helsinki, Finland) | |
| PBLH | 3-hour | Global Data Assimilation System (ftp://arlftp.arlhq.noaa.gov/pub/archives/gdas1) | PBLH at the sampling site was obtained using linear interpolation method. |

[*]WS, WD, T, P, DP, and PBLH represent wind speed, wind direction, temperature, pressure, dew point,
and planetary boundary layer height, respectively.

**Table S2.** Summary of output indices from the constructed $b_{ext}$ GAM.

| Intercept | 6.64 | |
|---|---|---|
| Adjusted $R^2$ | 0.54 | |
| Smoothed parameters[*] | F value | p value |
| $f$(WS) | 3.402 | 0.002331 |
| $f$(WD) | 5.820 | 0.000134 |
| $f$(T) | 2.707 | 0.012809 |
| $f$(P) | 3.209 | 0.001757 |
| $f$(DP) | 13.325 | $< 2.00 \times 10^{-16}$ |
| $f$(PBLH) | 3.656 | 0.026822 |

[*]WS, WD, T, P, DP, and PBLH represent wind speed, wind direction, temperature, pressure, dew point, and planetary boundary layer height, respectively.

**Table S3.** Concurvity indices between each independent smoothed parameter in the constructed GAM.

| Smoothed parameters[*] | $f$(WS) | $f$(WD) | $f$(T) | $f$(P) | $f$(DP) | $f$(PBLH) |
|---|---|---|---|---|---|---|
| $f$(WS) | 1.00 | 0.28 | 0.03 | 0.09 | 0.07 | 0.23 |
| $f$(WD) | 0.15 | 1.00 | 0.08 | 0.09 | 0.03 | 0.07 |
| $f$(T) | 0.06 | 0.07 | 1.00 | 0.11 | 0.25 | 0.22 |
| $f$(P) | 0.08 | 0.24 | 0.08 | 1.00 | 0.06 | 0.09 |
| $f$(DP) | 0.05 | 0.06 | 0.08 | 0.07 | 1.00 | 0.05 |
| $f$(PBLH) | 0.13 | 0.07 | 0.05 | 0.04 | 0.06 | 1.00 |

[*]WS, WD, T, P, DP, and PBLH represent wind speed, wind direction, temperature, pressure, dew point, and planetary boundary layer height, respectively.